# Tissue-resident macrophage and dendritic cells drive type I IFN immunity to enteroviruses in the liver

Emma Heckenberg[1], Jacob G. Davis[2], Caitlin Hale[2], Carolyn B. Coyne [1,2,3]*

**1** Department of Molecular Genetics and Microbiology, Duke University Medical School, Durham, North Carolina, United States of America, **2** Department of Integrated Immunobiology, Duke University Medical School, Durham, North Carolina, United States of America, **3** Duke Human Vaccine Institute, Duke University Medical School, Durham, North Carolina, United States of America

* carolyn.coyne@duke.edu

## Abstract

Enteroviruses are major causes of neonatal morbidity and mortality, with echovirus infections commonly associated with severe disease, including acute liver failure. The human neonatal Fc receptor (hFcRn) is the primary receptor for echoviruses, and its expression is required for infection of the liver in mouse models. While type I interferons (IFNs) are known to protect against echovirus-induced disease, the specific innate immune cells responsible for initiating this antiviral signaling in the liver remain undefined. To dissect the relative contributions of type I and type III IFNs in protecting the liver during echovirus infection, we combined *in vivo* mouse models (expressing hFcRn and deficient in Ifnar1, Ifnlr1, or both) with single cell RNA sequencing (scRNA-seq). This approach enabled us to pinpoint the hepatic cell types targeted by echoviruses and to identify the specific cells producing IFNs in response. We found that hepatocytes and Kupffer cells were the most heavily infected cell types. In contrast, early and robust type I IFN responses were primarily driven by Kupffer cells and a subset of dendritic cells. To determine whether type I IFNs act directly on hepatocytes to mediate protection, we generated conditional knockout mice lacking Ifnar1 specifically in hepatocytes. These mice showed similar morbidity, mortality, and hepatic viral titers as whole-body Ifnar1$^{-/-}$ animals, indicating that hepatocytes depend on protective IFN signals produced by immune cells during echovirus infection. These findings uncover cell-type-specific mechanisms by which echoviruses subvert host immunity and show how dysregulated IFN responses drive liver pathology and neonatal mortality.

## Author summary

Enteroviruses are a leading cause of severe illness and death in newborns, where acute liver failure is a prominent and life-threatening manifestation.

**PLOS Pathogens**

**Data availability statement:** Additional analysis code is available on the CoyneLab GitHub repository (https://github.com/CoyneLabDuke/Echovirus-infected-neonatal-mouse-livers). All raw files have been deposited on SRA under the following BioProjectID: PRJNA1303135.

**Funding:** This project was supported by NIH R01-AI150151 (C.B.C). The funders had no role in study design, data collection and analysis, decision to publish, or preparation of the manuscript.

**Competing interests:** The authors have declared that no competing interests exist.

However, the hepatic cell types that detect infection and initiate antiviral immunity remain largely undefined. Using mouse models that express the human neonatal Fc receptor (FcRn), the primary receptor for echoviruses, we define how the neonatal liver orchestrates an antiviral response at the single cell level. Leveraging single-cell RNA sequencing, interferon (IFN receptor knockout models, and a conditional knockout mouse model), we identify Kupffer cells, the liver's resident macrophages, along with a subset of dendritic cells, as the principal producers of type I IFNs during echovirus infection. These innate immune cells generate protective paracrine signals that enable hepatocytes, the liver's main metabolic cells, to restrict viral replication. Disruption of this IFN signaling axis results in uncontrolled viral spread, hepatocellular injury, and mortality. Together, our findings reveal how coordinated signaling between tissue-resident immune cells and hepatocytes protect the neonatal liver from enteroviral infection and provide insights into mechanisms underlying the unique vulnerability of newborns to echovirus-associated liver failure.

## Introduction

Neonatal viral infections remain a significant cause of morbidity and mortality worldwide, with enteroviruses representing a major contributor to this burden [1,2]. In the United States, enterovirus infections account for 33–65% of infant hospitalizations, with seasonal peaks during summer and autumn [3,4]. Among these, echoviruses are the most common in neonatal populations, with clinical outcomes ranging from mild febrile illness to acute liver failure and meningitis [5–8]. Outbreaks in neonatal intensive care units are particularly devastating, with mortality rates reaching one in three neonates, most often due to acute liver failure [9,10]. A recent outbreak across Europe (2022–2023) saw case fatality rates rise to 78%, underscoring the persistent and evolving threat echoviruses pose to vulnerable neonatal populations [11].

Echoviruses are transmitted via the fecal-oral route and are typically controlled by the innate immune system [12]. However, in severe cases, the virus can disseminate hematogenously to secondary organs such as the liver [13–15]. Despite the known hepatotropism of echoviruses, the specific cell populations responsible for initiating antiviral responses in the liver remain poorly defined. During acute RNA virus infections, the liver relies primarily on innate immune cells including Kupffer cells (KCs), dendritic cells (DCs), and neutrophils to mount a type I interferon (IFN) response [16]. In contrast, chronic RNA viruses often evade type I IFN responses and establish infection directly in hepatocytes. In these settings, viral control is more often associated with type III IFN production by hepatocytes themselves [17–19]. Type III IFNs signal through a distinct receptor complex primarily expressed on epithelial and barrier cells, including hepatocytes, and are thought to promote localized antiviral defense while minimizing tissue inflammation [20].

Among innate immune cells, KCs are the liver's resident macrophages and play a central role in the sensing of blood-borne pathogens and initiating inflammatory

signaling [21,22]. In the liver, DCs interact closely with KCs and hepatocytes to sample antigens and modulate immune responses [23]. During infection, neutrophils are rapidly recruited to the liver where they contribute to inflammation and aid in tissue repair by clearing necrotic debris and releasing pro-resolving mediators [23]. Previously, we demonstrated that echovirus infection in mice lacking the receptor for type I IFNs (Ifnar1⁻ᐟ) leads to robust hepatocyte infection [24], suggesting that hepatocytes are important targets of viral replication and may be the primary sensors of antiviral IFNs. However, despite the clinical relevance of echovirus infections in neonates, most experimental models of enteroviral hepatitis have focused on adult animals. During early postnatal development, the liver is still undergoing structural maturation, including vascular remodeling and zonation, and has a distinct balance of immune cell populations, with reduced numbers of fully mature antigen-presenting cells and limited effector lymphocyte recruitment [25–28]. Moreover, both the production and responsiveness to type I IFNs are attenuated in neonates compared to adults [29]. These developmental differences may fundamentally alter how the neonatal liver detects, amplifies, and responds to enteroviral infections. As a result, how the immature liver coordinates innate antiviral immunity during this critical window of susceptibility remains poorly understood.

To address these gaps, we systematically defined the cellular landscape of the neonatal liver during echovirus infection, with a focus on identifying the sources and targets of IFN signaling across key immune and parenchymal cell types. To do this, we generated a comprehensive single-cell (sc) RNA-seq dataset from echovirus-infected livers of neonatal Tg32 mice, which express human FcRn (hFcRn), the primary receptor for echoviruses [30]. These mice represented distinct IFN signaling capacities: immunocompetent (Ifnar1$^{+/+}$Ifnlr1$^{+/+}$), Ifnar1⁻ᐟ⁻ (deficient in type I IFN signaling), Ifnlr1⁻ᐟ⁻ (deficient in type III IFN signaling), and a newly generated Ifnar1⁻ᐟ⁻Ifnlr1⁻ᐟ⁻ double knockout strain lacking both pathways. This approach enabled IFN receptor-specific resolution of viral tropism and host responses. We found that KCs and a subset of conventional DCs (cDCs) were the primary producers of type I IFNs in the infected liver, with a small population of neutrophils participating to a smaller extent. KCs not only exhibited the highest levels of *Ifnb1* expression, but also underwent a striking expansion during infection, distinguishing them from other myeloid populations. Notably, KCs also harbored the highest viral loads, revealing a tight link between viral sensing and IFN production. To test whether liver immune cell-intrinsic IFN signaling is required for protection, we generated conditional Ifnar1 knockout mice lacking type I IFN receptors specifically in hepatocytes. These mice exhibited hepatic pathology and viral titers comparable to whole-body Ifnar1⁻ᐟ⁻ animals, demonstrating that hepatocytes depend on type I IFNs produced by immune cells to control infection. These findings uncover cell-type-specific mechanisms of immune defense and viral evasion in the neonatal liver, with important implications for understanding neonatal hepatitis, the development of liver immunity, and the pathogenesis of enterovirus-associated mortality.

## Results

### Loss of type I IFN signaling drives echovirus-associated mortality and liver infection

To investigate how type I and type III IFNs differentially protect the neonatal liver from echovirus-induced disease, we employed a panel of four transgenic mouse models to capture early acute infection. Building on our prior work showing that adult hFcRn-expressing mice lacking type I IFN signaling (hFcRn$^{Tg32}$ Ifnar1⁻ᐟ) are highly susceptible to echovirus infection with hepatocyte-specific viral tropism [31], we extended our analysis to include mice lacking type III IFN signaling (hFcRn$^{Tg32}$ Ifnlr1⁻ᐟ⁻), as well as a newly generated double knockout strain lacking both type I and type III IFN receptors (hFcRn$^{Tg32}$ Ifnar1⁻ᐟ⁻Ifnlr1⁻ᐟ⁻). For simplicity, we refer to these strains hereafter as Ifnar1⁻ᐟ⁻ Ifnlr1⁻ᐟ⁻, and Ifnar1⁻ᐟ⁻Ifnlr1⁻ᐟ⁻. Neonatal pups were infected intraperitoneally with echovirus 5 (E5), a representative strain used due to its reliance, exclusively, on hFcRn for infection and not DAF as an additional attachment factor [32]. Based on our prior findings that loss of type I IFN signaling results in dramatically increased mortality [24,31], we optimized a new infection model in which Ifnar1⁻ᐟ⁻ and Ifnar1⁻ᐟ⁻Ifnlr1⁻ᐟ⁻ mice were infected with 50 PFU of E5, whereas wild-type (Wt), and Ifnlr1⁻ᐟ⁻ mice received a higher dose of 10⁶ PFU. As expected, Ifnar deficient strains (Ifnar1⁻ᐟ⁻ and Ifnar1⁻ᐟ⁻Ifnlr1⁻ᐟ⁻) succumbed to infection by 96 hours post infection (hpi) whereas Wt and Ifnlr1⁻ᐟ⁻, mice survived beyond this early acute timepoint (Fig 1A). To capture which liver cell

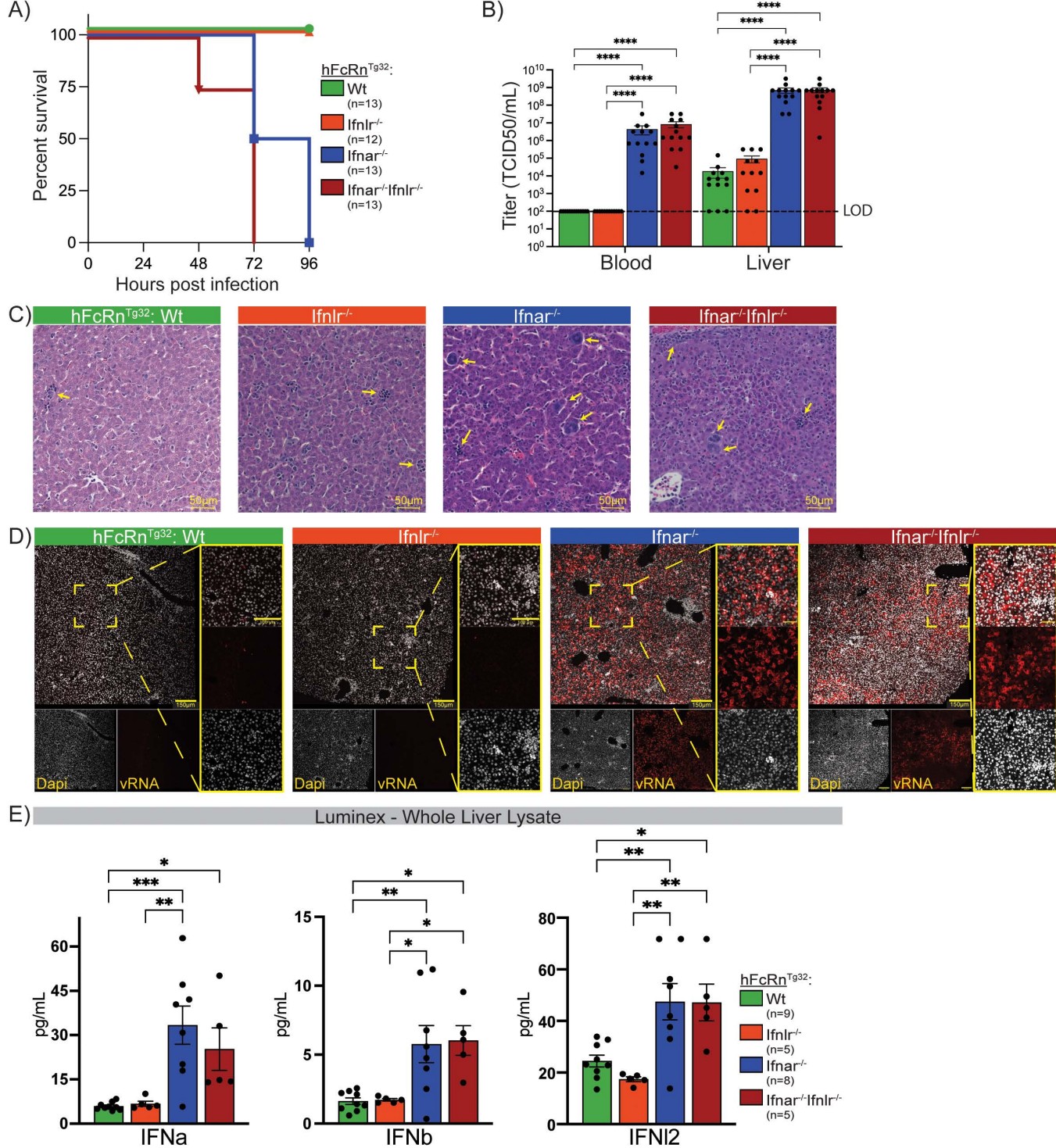

**Fig 1. Type I IFN deficiency increases viral burden and mortality during echovirus infection. A)** Early acute survival curve of four interferon knockout neonatal mouse models infected with echovirus. All mice are on a human neonatal Fc receptor background (hFcRcn$^{Tg32}$): green line represents immunocompetent (Wt) genotype, orange lacks the type III IFN receptor (hFcRcn$^{Tg32}$Ifnlr1$^{-/-}$), blue lacks the type I IFN receptor (hFcRcn$^{Tg32}$ Ifnar1$^{-/-}$), and the red line represents the double knockout genotype which lacks both type I and type III IFN receptors (hFcRcn$^{Tg32}$ Ifnar1$^{-/-}$ Ifnlr1$^{-/-}$). Wt and Ifnlr1$^{-/-}$pups were challenged with 10$^6$pfu of echovirus5 (E5) via intraperitoneal (IP) infection. Ifnar1$^{-/-}$ and Ifnar1$^{-/-}$ Ifnlr1$^{-/-}$ pups were challenged with 50pfu of E5 via

an IP route. **B)** Viral titers from blood and liver of infected mice, collected 40hpi. Asterisks represent samples with statistically significant differences in viral titers across genotypes as determined by one-way ANOVA tests (p < 0.0001). Limit of detection for the assay is marked by a dashed black line. **C)** H&E images (scale bar, 50µm) of infected livers at 40hpi across all four genotypes. Ifnar1$^{-/-}$ and Ifnar1$^{-/-}$ Ifnlr1$^{-/-}$ mice have more immune cell infiltrates and hepatocyte damage than Wt and Ifnlr1$^{-/-}$ pups as indicated by yellow arrows. **D)** Confocal microscopy images (scale bars, 150µm and 100µm) of echovirus RNA in infected neonatal mouse livers. Hybridization chain reaction (HCR) was used to detect E5 RNA (red) in liver sections from neonatal mice at 40hpi. Ifnar1$^{-/-}$ and Ifnar1$^{-/-}$ Ifnlr1$^{-/-}$ livers showed higher levels of E5 RNA compared to wild-type and Ifnlr1$^{-/-}$ livers at the same time point. Nuclei are stained with DAPI (grey). **E)** Luminex protein quantification of type I (Ifnα and Ifnβ) and III (Ifnl2) IFNs from whole liver lysate – same samples used for titer data generation. Values are shown as pg/mL. Asterisks represent samples with statistically significant differences in viral titers across genotypes as determined by one-way ANOVA tests (* p < 0.05, ** p < 0.005, *** p < 0.0005).

types were participating in the early-acute response to echovirus infection we selected a 40-hour harvest timepoint. At this stage, animals across all four genotypes remained alive, allowing direct comparison of innate immune activity among models. Consistent with survival dynamics, viral titers at 40hpi in the livers and blood of Ifnar1 deficient strains were significantly higher than those observed in Wt or Ifnlr1$^{-/-}$ strains (Fig 1B). Viral titers did not differ between sex across any of the genotypes tested (S1A Fig). Notably, viral titers were comparable between Ifnar1$^{-/-}$ and Ifnar1$^{-/-}$ Ifnlr1$^{-/-}$ mice, indicating that type III IFN signaling does not provide additional protection in the absence of type I IFNs during echovirus infection.

We next determined the impact of E5 infection on liver structure and integrity across genotypes. Using hematoxylin and eosin (H&E) staining we observed robust immune cell infiltration and hepatocyte destruction in Ifnar1$^{-/-}$ and Ifnar1$^{-/-}$ Ifnlr1$^{-/-}$ models at 40hpi following echovirus infection (Fig 1C). In contrast, infected Wt and Ifnlr1$^{-/-}$ livers were comparable to uninfected controls (S1B Fig). Finally, we used hybridized chain reaction (HCR) to localize echovirus replication in the liver. Consistent with viral titer data, we observed very low viral RNA in Wt and Ifnlr1$^{-/-}$ strains (Fig 1D). However, there was high viral RNA burden in both the Ifnar1$^{-/-}$ and Ifnar1$^{-/-}$ Ifnlr1$^{-/-}$ strains.

Finally, to quantify and characterize the IFN response across genotypes during early-acute echovirus infection, we performed Luminex protein analysis on whole liver lysates from the same samples used to generate titer data. All genotypes induced both type I (Ifnα and Ifnβ) and type III (Ifnl2) IFNs (Fig 1E). IFN levels were significantly higher in Ifnar$^{-/-}$ mice compared with Ifnar$^{+/+}$ controls, reflecting the increased viral replication and consequent accumulation of pathogen-associated molecular patterns (PAMPs) that drive heightened IFN induction in the absence of functional signaling. Together, these findings demonstrate that type I, but not type III, IFN signaling is essential for survival during echovirus infection and for limiting viral replication in the liver.

## Single-cell transcriptomics defines echovirus tropism and IFN responses in the liver

To comprehensively define liver cell types susceptible to echovirus infection and characterize their transcriptional responses, we performed scRNA-seq on the livers from neonatal mice across all four genotypes (Wt, Ifnlr1$^{-/-}$, Ifnar1$^{-/-}$, and Ifnar1$^{-/-}$ Ifnlr1$^{-/-}$). For each genotype, we collected livers from three uninfected and three E5-infected littermates at 40 hpi, resulting in eight experimental conditions. Livers were manually dissociated into single-cell suspensions, and samples from each mouse were processed and sequenced individually. Reads were mapped to a custom reference genome constructed according to the 10X Genomics assembly rules [33]. The E5 genome and the human neonatal Fc receptor (hFcRn) were added as extra chromosomes to the mm10 reference genome from 10X Genomics [34]. A total of 219,980 cells passed quality control thresholds and were visualized using UMAP projections (Fig 2A). To account for technical variation between samples, we applied Harmony for batch correction during integration. In addition, sex, number of detected features (nFeature_RNA), total UMI counts (nCount_RNA), and percentage of mitochondrial transcripts (%mt) were regressed out during data normalization. We then examined cluster composition across genotypes and infection conditions (Fig 2B). Cluster proportions and total cell numbers did not vary significantly by infection status, individual mouse, or sex (S2A, S2B, and S2D Fig), confirming the robustness of the dataset for downstream analyses.

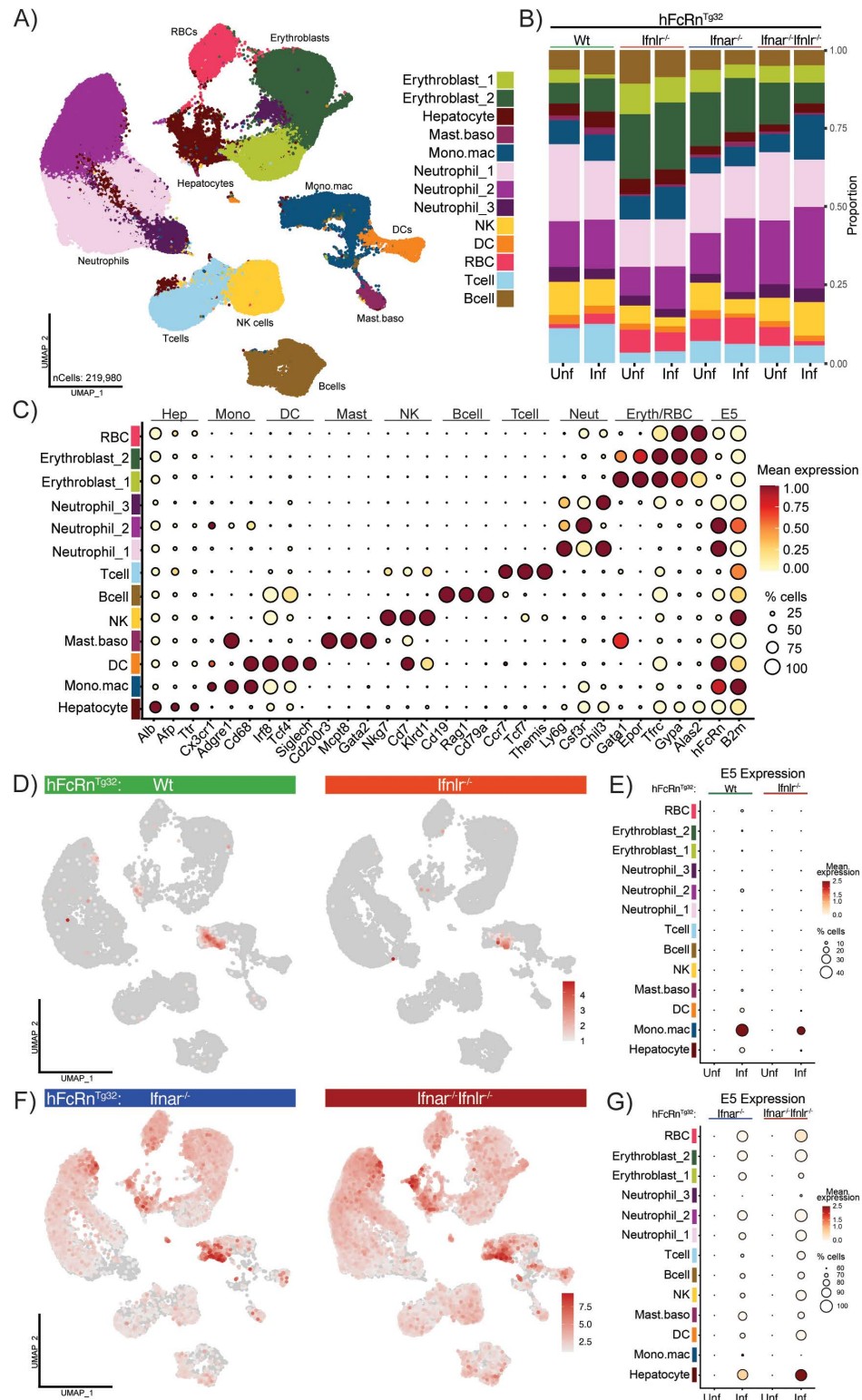

**Fig 2. Loss of type I IFN signaling broadens echovirus tropism in the neonatal liver. A)** UMAP of integrated liver datasets from all four IFN knock-out models at an early acute timepoint (40hpi). Each genotype is represented by three uninfected and three infected littermates for a total of 8 individual datasets. Cell types are annotated as follows: erythroblasts (greens), hepatocytes (dark maroon), mast cell and basophils (mast.baso- light maroon),

monocytes and macrophages (mono.mac- dark blue), neutrophils (purples), natural killer cells (NK- yellow), dendritic cells (DC- orange), red blood cells (RBC- red), T-cells (light blue), and B-cells (brown). Cell types that contained multiple clusters are identified by "_number" after the name. **B)** Bar plot demonstrating the proportion of each cell cluster within the dataset, split by genotype (top) and infection status (bottom). **C)** Dot plot of key markers for each cell type in the combined dataset, E5 receptors (hFcRn and B2m) are included. Values are scaled with 1 (red) representing the highest, normalized, expression of a given gene and 0 (white) representing the lowest normalized expression. Dot size indicates the percentage of total cells in a cluster that express the gene of interest. **D)** Feature plots of normalized expression of E5 transcript (red) in infected Wt and Ifnlr1$^{-/-}$ genotypes across the whole liver. **E)** Dot plots of scaled E5 transcript in uninfected and infected Wt and Ifnlr1$^{-/-}$ genotypes. E5 transcript maps strongly to the Mono.mac cluster in both genotypes as indicated by high mean expression (red) and the percent cells that have transcript E5 mapping to them (dot size). **F)** Feature plots of E5 transcript (red) in infected Ifnar1$^{-/-}$ and Ifnar1$^{-/-}$ Ifnlr1$^{-/-}$ genotypes across the whole liver. **G)** Dot plots of scaled E5 transcript in uninfected and infected Ifnar1$^{-/-}$ and Ifnar1$^{-/-}$ Ifnlr1$^{-/-}$ genotypes. E5 tropism in the liver expands in the absence of *Ifnar1* expression as highlighted the percentage of cells in each cluster that contain mapped E5 transcript (dot size).

We identified 10 distinct cell types across the 13 transcriptional clusters, based on expression of established gene markers (Fig 2C and S1 Table) [22,25,28,35,36]. Among these clusters, we detected well-characterized liver-resident populations, including hepatocytes (high expression of *Alb*, *Afp*, and *Ttr*) and a heterogeneous monocyte/macrophage cluster (high expression of *Cd68*, *Cx3cr1*, and *Adgre1*). We also identified multiple populations of circulating professional immune cells, including dendritic cells (*Siglech*, *Tcf4*, and *Irf8* high), a mixed mast cell/basophil population (*Mcpt8*, *Gata2*, and *Cd200r3* high), a natural killer (NK) cell population (*Nkg7*, *Cd7*, and *Klrd1*), B-cells (*Cd19*, *Rag1*, and *Cd79a*), and T-cells (*Ccr7*, *Tcf7*, and *Themis* high). In addition, three transcriptionally distinct neutrophil clusters were identified (marked by *Ly6g*, *Csf3r*, and *Chil3*). Finally, three of the 13 clusters represented erythroid lineage cells, including red blood cells (*Alas2*, *Gypa*) and erythroblasts (*Gata1*, *Epor*, *Tfrc*). Given the diversity of identified cell types and the consistent representation of clusters across genotypes and conditions, we next focused on determining which of these cell types harbored echovirus transcripts.

While hFcRn has been detected in various liver cell types in both human tissue and transgenic mouse models, its expression at single-cell resolution remains poorly defined. Notably, IHC analysis in Tg32 mice demonstrates an hFcRn expression pattern similar to that observed in human liver, including localization to Kupffer cells and hepatocytes [37]. To identify candidate cell types likely to support echovirus entry, we analyzed the expression of the two genes encoding the echovirus receptor heterodimer: *hFcRn* and beta-2-microglobulin (*B2m*), across the dataset. Six clusters showed high expression of both genes including hepatocytes, monocytes/macrophages, dendritic cells, mast cells/basophils, and two neutrophil populations (Neutrophil_1 and Neutrophil_2) (Fig 2C). These clusters were thus predicted to be the most susceptible to echovirus infection. We next examined the presence of E5 viral RNA across the datasets to determine how infection varied by IFN signaling status. Notably, this dataset allows us to identify cell types that harbor echovirus RNA but does not distinguish whether individual cells are productively infected. However, when integrated with viral titer data (Fig 1B), the results indicate that the livers of these mice are productively infected overall. To localize viral RNA, we separated the dataset by genotype and analyzed only the infected samples. In Wt and Ifnlr1$^{-/-}$ mice, viral reads were primarily detected in the monocyte/macrophage (Mono.mac) cluster, with lower-level expression also observed in hepatocytes (Figs 2D–2E and S2C). These findings are consistent with survival and viral titer data, indicating that intact type I IFN signaling restricts viral spread. In contrast, loss of type I IFN signaling markedly expanded the range of infected cell types. In both Ifnar1$^{-/-}$ and Ifnar1$^{-/-}$ Ifnlr1$^{-/-}$ mice, echovirus RNA was detected broadly across nearly all clusters. Despite this widespread dissemination, the hepatocyte and Mono.mac clusters still exhibited the highest levels of viral RNA by mean gene expression (Figs 2F, 2G and S2E), highlighting their role as major targets of infection in the absence of type I IFN-mediated control. These data demonstrate that type I IFN signaling is essential for restricting echovirus infection to a narrow set of target cells in the liver, and that in its absence, both viral tropism and burden expand dramatically, particularly within hepatocytes and monocyte/macrophage populations.

## Coordinated IFN signaling by myeloid cells shapes the hepatic inflammatory response to echovirus

To control damage during early acute virus infections, the liver canonically induces a broad IFN response to defend against invading pathogens [38]. However, the specific cell types in the liver that are responsible for producing an IFN

response during echovirus infection are unknown. To address this, we next examined how distinct cellular clusters activated and upregulated inflammatory and antiviral pathways in response to infection. We first analyzed the induction of IFNs (type I, type II, or type III) between infected and uninfected samples across all four genotypes using normalized gene counts (S2 Table). To visualize these patterns, we generated heatmaps of $\log_2$-transformed expression values across the eight conditions. We found that *Ifna2*, *Ifnb1*, and *Ifnl2* were each induced to varying degrees in response to infection (Fig 3A–3C). Among these, *Ifnb1* stood out for its robust induction across all four genotypes, particularly in key cell populations including hepatocytes, monocyte/macrophages, dendritic cells, and neutrophils (Fig 3B).

Next, to assess how antiviral IFNs (type I and type III) shape the inflammatory landscape of the liver across genotypes, we calculated an IFN-stimulated gene (ISG) score using Seurat's package *AddModuleScore*. This approach computes the average expression of a predefined gene set and subtract background expression from control genes, enabling a direct comparison of ISG expression across cell populations and genotypes. To generate a list of candidate ISGs, we performed differential gene expression analysis using DeSeq2. We first identified genes that were significantly differentially expressed between Wt-infected and Wt-uninfected livers. To focus on the most robust changes, we applied a cutoff of $\log_2$ fold change > 4 and $p < 0.05$. The same criteria were then used to identify significantly differentially expressed genes between Wt-infected and Ifnar1$^{-/-}$ Ifnlr1$^{-/-}$-infected livers (S3 Table). By intersecting these two gene sets, we identified 11 shared ISGs that met our criteria: *Ifit3*, *Ifit3b*, *Siglec1*, *Apol9b*, *Ifi44*, *Ifit1*, *Oas1g*, *Apod*, *Phf11d*, *Sct,* and *Ifit1bl1*. This final gene set represents antiviral IFN dependent genes that are induced during echovirus infection and was used to calculate an ISG score, which we visualized using FeaturePlots across infected and uninfected conditions in all genotypes (S3 Table). Consistent with the survival and low titers observed in Wt and Ifnlr1$^{-/-}$ pups, these genotypes mounted a robust inflammatory response to echovirus infection, as indicated by elevated ISG scores (Fig 3D, 3E, 3H, 3I). In contrast, the inflammatory response was markedly diminished in Ifnar1$^{-/-}$ and Ifnar1$^{-/-}$ Ifnlr1$^{-/-}$ livers, with the notable exception of neutrophil clusters in the Ifnar1$^{-/-}$ samples, which retained partial ISG induction (Fig 3F, 3G, 3J, 3K).

To assess how Ifnar ablation affected the inflammatory profile of clusters identified as type I IFN producers, we generated dot plots displaying expression of the 17 ISGs used to calculate the ISG score. In the Mono.mac and DC clusters, Ifnar ablation impaired the induction of 13 of the 17 ISGs in response to infection (Fig 3L, 3M). In contrast, Wt and Ifnlr1$^{-/-}$ mice exhibited strong induction of all 17 ISGs. Hepatocytes are known to heavily rely on type III IFN signaling during viral infection [18,39], a pattern reflected in their ISG induction following echovirus infection. Consistent with this, Wt, Ifnlr1$^{-/-}$, and Ifnar1$^{-/-}$ hepatocytes each upregulated antiviral ISGs in response to infection (Fig 3N). In contrast, hepatocytes from Ifnar1$^{-/-}$ Ifnlr1$^{-/-}$ mice failed to upregulate ISGs to the same extent, indicating that the antiviral response in this population is entirely dependent on IFN signaling. Together, these analyses identified four major clusters (Mono.mac, DC, hepatocytes, and neutrophils) that upregulated type I IFNs in response to echovirus infection across the four genotypes. They also demonstrate that type I IFNs are essential for driving a protective inflammatory response, particularly in Ifnar1$^{-/-}$ models Given the cellular heterogeneity within these broad clusters, we next sought to resolve which specific subpopulations are responsible for IFN production across these four major clusters. Because the Mono.mac cluster harbored the highest number of viral reads in immunocompetent genotypes, we begin our subcluster analysis there, before moving on to the dendritic cell, hepatocyte, and neutrophil clusters.

## Kupffer cells expand and serve as the primary source of type I IFNs during echovirus infection

To improve resolution and better define the cellular composition of key clusters across the eight experimental conditions, we performed sub-clustering of select populations. We began by focusing on the monocyte/macrophage (Mono.mac) cluster due to its robust type I IFN production and ISG responsiveness. To improve resolution, we extracted all Mono.mac cells across the eight experimental conditions and re-performed integration and clustering. To ensure the specificity of this subset, we screened for expression of lineage-defining markers (*Adgre1*, *Cd68*, *Siglech*, *Cd200r3*, *Nkg7*, *Gata1*) and removed three contaminating clusters identified as NK cells, a mast cell/basophil mix, and a DC population. We then

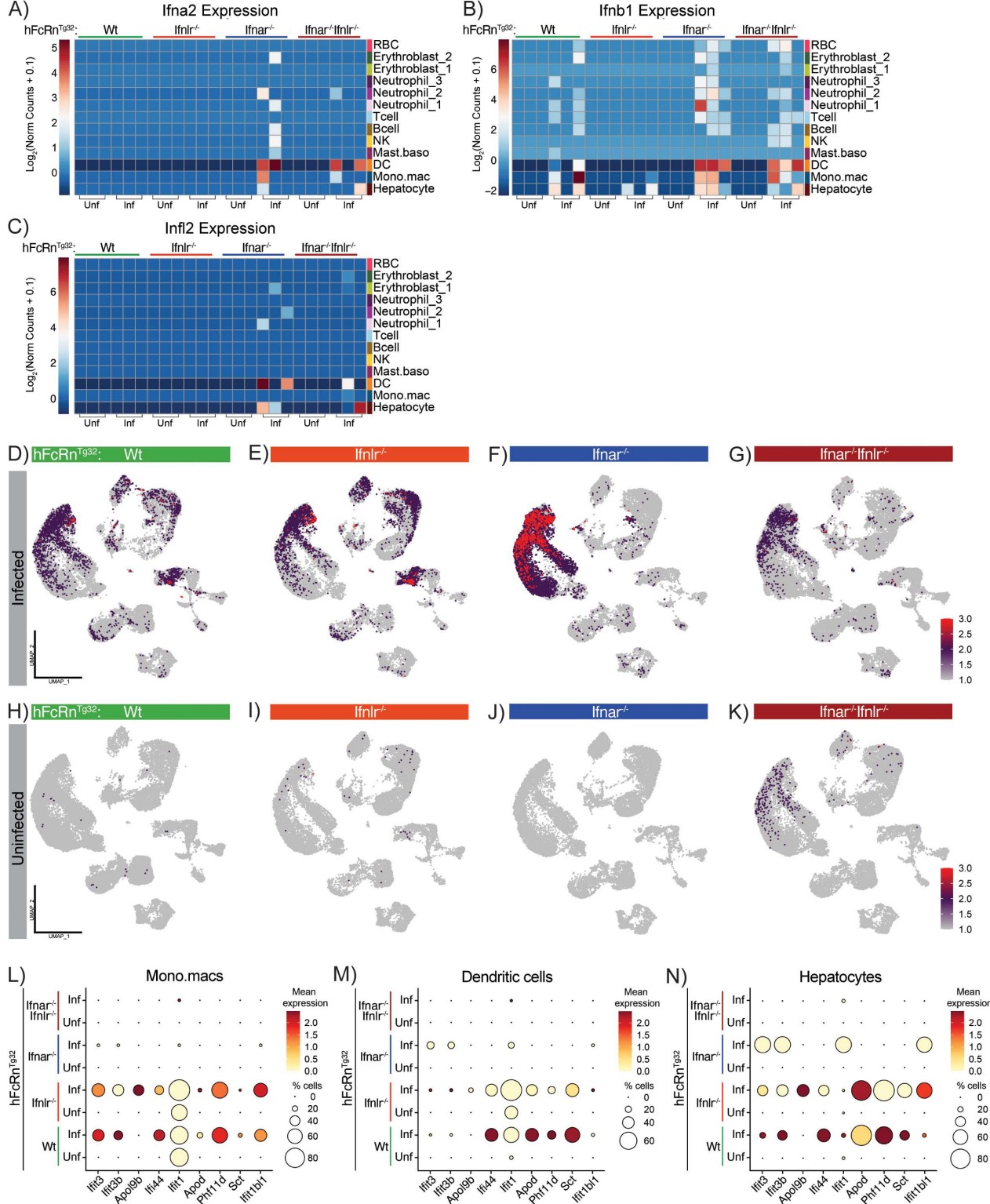

**Fig 3. Myeloid cell–driven interferon responses mediate inflammation during acute echovirus infection. A-C)** Heatmap to visualize *Ifna2, Ifnb1,* and *Ifnl2* transcript from each sample, grouped by infection status (x-axis), genotype (top) and cell type (right). Values represent the Log₂ (normalized counts) of the transcripts for each IFN with red indicating the highest expression of each gene in the dataset and dark blue reflecting no expression.

**D-G)** Feature plots mapping the ISG score results to all cell types across infected Wt, Ifnlr1$^{-/-}$, Ifnar1$^{-/-}$, and Ifnar1$^{-/-}$ Ifnlr1$^{-/-}$ samples. Red dots represent the highest expression of the ISG score in a cell, purple reflects a moderate expression of the score, and grey reflects no expression. The ISG score was generated using the *AddModuleScore* package applied to a panel of 11 ISGs (*Ifit3, Ifit3b, Siglec1, Apol9b, Ifi44, Ifit1, Oas1g, Apod, Phf11d, Sct,* and *Ifit1bl1*) differentially expressed between both uninfected and infected Wt conditions and infected Wt vs infected Ifnar1$^{-/-}$ conditions. **H-K)** Feature plots mapping the ISG score results to all cell types across uninfected Wt, Ifnlr1$^{-/-}$, Ifnar1$^{-/-}$, and Ifnar1$^{-/-}$ Ifnlr1$^{-/-}$ samples. Gene expression scales match the infected samples above. **L-N)** Dot plots of mean RNA expression of each of the individual ISGs that contribute to the ISG score across key IFN producing cell types (Mono.macs, DCs, hepatocytes, respectively). Dot plots are grouped by genotypes and infection status (y-axis) with mean expression and percent cells for each cluster as the legend on the right.

re-ran clustering on the dataset, yielding 16,527 cells (Fig 4A). Based on canonical monocyte and macrophage markers [22,36], we identified six distinct cell types: proliferating monocytes (*Top2a* high), classic monocytes (*Ccr2* and *Ly6c2* high), transitioning monocyte.macrophages (*Ccr2*, *Adgre1*, *Cd68* high), patrolling monocyte- macrophages (*Spn*, *Itgam* high), inflammatory macrophages (*Card11* high), and liver resident Kupffer cells (*Marco*, *Clec4f*, *Folr2*, *Timd4* high) (Fig 4B and S4 Table). Expression of *hFcRn* and *B2m*, which are necessary for echovirus infection, were highly expressed across all identified clusters, highlighting the potential for multiple liver cell types to serve as viral targets (Fig 4D). All six monocyte/macrophage subtypes were present across the eight experimental conditions. However, when we examined changes in cell type proportions, we found that the KC cluster uniquely expanded in response to infection, showing increased abundance across all four genotypes (Figs 4C, 4F, and S3A and S5 Table). This expansion was not observed in any other monocyte or macrophage subset and was specific to infected conditions. Together, these results implicate Kupffer cells as key responders to echovirus infection in the liver.

To confirm that the expanding Kupffer cell cluster represented a distinct population from the infiltrating inflammatory macrophage cluster, we performed trajectory analysis on the Mono.mac dataset using Slingshot. This approach infers a pseudotemporal ordering of cells, allowing us to model how cell types mature or transition over time. Three trajectories were predicted using the Slingshot algorithm, all originating from the Prolif.monocytes cluster (Figs 4E and S3B). Trajectory analysis revealed three distinct developmental pathways: one terminating in the Kupffer cell cluster (top right panel), a second terminating in the inflammatory macrophage (Inflam.mac) cluster (bottom right panel), and a third looping back to the proliferating monocyte cluster (S3C Fig, right panel). The separation of Kupffer cells and inflammatory macrophages into distinct trajectories suggests that the expansion of the Kupffer cell cluster during infection is consistent with differentiation from circulating monocyte precursors, rather than reflecting a direct transition from inflammatory macrophages into the resident Kupffer cell niche, in line with previous descriptions of monocyte-derived Kupffer cell replenishment in the liver.

After identifying the Mono.mac cluster as a major source of type I IFNs at the whole-liver level, we next sought to pinpoint which specific cell types within this heterogeneous population were responsible for IFN induction during echovirus infection. To do this, we examined *Ifnb1* expression as a representative type I IFN. When we mapped *Ifnb1* transcripts across infected and uninfected samples from all genotypes, we found that Kupffer cells were the primary source of *Ifnb1* expression in Wt, Ifnar1$^{-/-}$, and Ifnar1$^{-/-}$Ifnlr1$^{-/-}$ mice, based on mean expression levels. (Figs 4G and S3C). Notably, Ifnar1$^{-/-}$ and Ifnar1$^{-/-}$Ifnlr1$^{-/-}$ genotypes showed broader *Ifnb1* induction across multiple Mono.mac subtypes, suggesting that in the absence of downstream IFN signaling, cells may sustain or amplify *Ifnb1* production in a compensatory attempt to control infection.

Next, we examined the distribution of echovirus transcripts across cell types and genotypes to determine whether IFN-producing cells were also sites of viral infection. Using Seurat's blend visualization to co-map echovirus and cell-type–specific markers, we found that in Wt and Ifnlr1$^{-/-}$ livers, viral transcripts localized predominantly to Kupffer cells, as indicated by co-expression with *Marco* (Figs 4H and S3D, S3E - left panel). In contrast, echovirus transcripts in Ifnar1$^{-/-}$ and Ifnar1$^{-/-}$Ifnlr1$^{-/-}$ mice were broadly distributed across multiple cell types, though they remained most concentrated in the Kupffer cell and inflammatory macrophage clusters (Figs 4H- right panel and S3D, S3E- right panel). To determine whether *Ifnb1*-producing

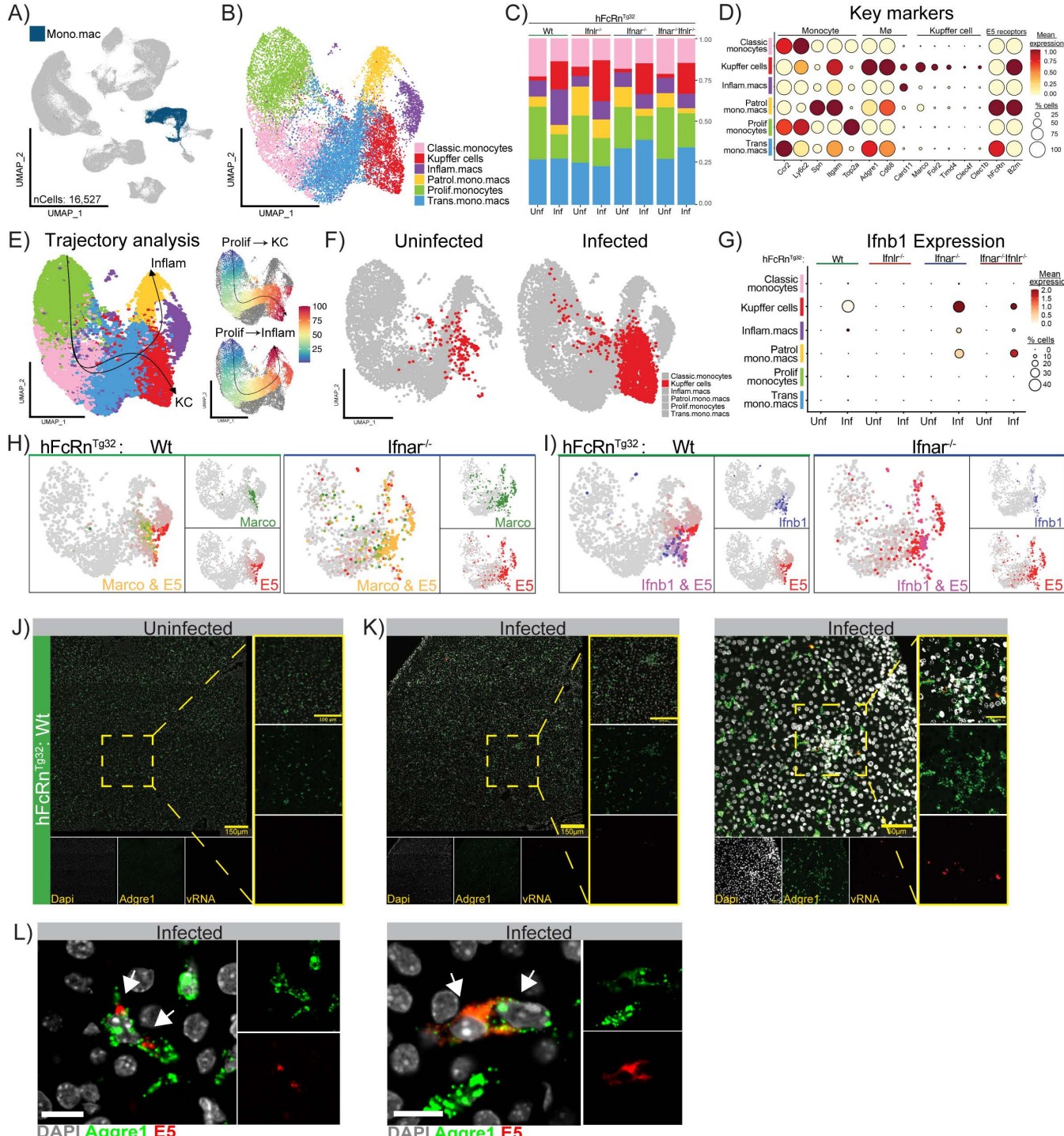

**Fig 4. Kupffer cells harbor echovirus and are the main producer of type I Ifns during infection. A)** UMAP highlighting the Mono.mac cluster in the original whole-liver dataset. This cluster is made up of 16,527 cells across all 8 experimental conditions. **B)** UMAP of the reclustered Mono.mac population, new cell identities were assigned based on key marker expression. Cell types are annotated as follows: Kupffer cells (red), patrolling monocytes and macrophages (Patrol.mono.macs, yellow), classic monocyte (pink), inflammatory macrophages (Inflam.macs, purple), transitioning monocytes and macrophages (Trans.mono.macs, blue), and proliferating monocytes (Prolif.monocytes, green). **C)** Bar plot showing the proportion of each cell cluster within the subsetted dataset, split by genotype (top) and infection status (bottom). **D)** Key markers (genes on x-axis) for each cell type (y-axis) including

E5 receptors are shown as a Dot plot. Scaled mean expression is shown as a color scale and the percent of cell expressing each gene is demonstrated by the size of the dot. Labels on top of the graph indicate gross groupings of key genes. **E)** Slingshot trajectories superimposed and labeled (Inflam or KC) on the reclustered Mono.mac UMAP (left panel). Feature plots of the pseudotime values (0-100) assigned to each cell along each trajectory (right panels). The color gradient (blue to red) and arrow indicate the directionality of cell differentiation along these pathways. **F)** UMAP of the reclustered Mono.mac population split by infection status to highlight the difference in abundance of the Kupffer cell cluster (red). **G)** Dot plot showing the average expression and proportion of cells expressing *Ifnb1* across the different cell types (y-axis) separated by genotype (top) and infection status (x-axis). **H)** Merged Feature plots of infected Wt (left) and Ifnar1$^{-/-}$ (right) samples demonstrating expression of a key Kupffer cell marker *Marco* (green) and E5 (red). Yellow indicates that a given cell expresses both *Marco* and E5. **I)** Merged Feature plots of infected Wt (left) and Ifnar1$^{-/-}$ (right) samples showing expression of *Ifnb1* (blue) and E5 (red). Purple indicates that a given cell co-expresses *Ifnb1* and E5. **J and K)** Confocal microscopy images (scale bars, 150 μm and 100 μm) showing expression of the Kupffer cell marker *Adgre1* and echovirus RNA in uninfected and infected Wt neonatal mouse livers, respectively. RNAscope was used to detect *Adgre1* (green) and E5 viral RNA (red) in liver sections from uninfected or 40 hpi mice. Uninfected livers display a uniform distribution of *Adgre1*$^+$ cells without detectable echovirus signal, whereas infected livers show *Adgre1*$^+$ Kupffer cells throughout the tissue, with localized "hotspots" of cells co-expressing E5 RNA. Nuclei are stained with DAPI (gray). **L)** 60x confocal images of the slides described above. Echovirus (E5) positive cells in red, Adgre1 (macrophage) positive cells are in green, nuclei are in grey (DAPI) Scale bars (10μm).

Kupffer cells were themselves infected, rather than merely responding to signals from neighboring cells, we mapped co-expression of echovirus transcripts (red) and *Ifnb1* (blue). In Wt and Ifnar1$^{-/-}$ mice, approximately 40% of virus-positive cells also expressed *Ifnb1* (137/336 and 81/217 cells, respectively; Fig 4I and S6 Table). In contrast, Ifnar1$^{-/-}$ Ifnlr1$^{-/-}$ mice exhibited a reduced frequency of *Ifnb1* induction, with only 20% (197/1,020) of virus-harboring cells expressing *Ifnb1* (S3F Fig).

Finally, to visualize echovirus localization relative to Kupffer cells in the liver, we performed RNAscope on liver sections from Wt pups. As expected, viral RNA was absent in uninfected Wt livers despite robust expression of the canonical Kupffer cell marker *Adgre1* (Fig 4J). In contrast, infected Wt livers showed colocalization of echovirus RNA with *Adgre1* in a small subset of liver-resident macrophages (Fig 4K and 4L), consistent with our scRNA-seq findings. Echovirus-positive cells that also expressed the macrophage marker Adgre1 were quantified using unbiased image analysis. Ten 40×fields containing echovirus-positive cells were randomly captured, and post-acquisition analysis was used to determine co-expression of Adgre1. Using this approach, we found that ~96% of echovirus-positive cells (24/25) in the livers of immuno-competent mice were also Adgre1-positive. This observation is consistent with our single-cell data. Together, these results suggest that Kupffer cells act as early responders during echovirus infection, directly sensing viral presence and initiating a type I interferon response to alert neighboring cells.

### Conventional dendritic cells and kupffer cells act in concert to induce type I IFN responses

In addition to the Mono.mac cluster, we identified a heterogeneous DC cluster that also induced a type I IFN response across all genotypes during echovirus infection. To investigate whether a distinct cell type within the DC-like population, similar to Kupffer cells, was responsible for viral sensing and IFN induction, we applied the same analytical approach described previously. First, we selected and subsetted the DC cluster from the whole liver dataset, which consisted of 4,622 cells across all eight conditions (Fig 5A). Next, we performed PCA and re-clustering on the DC population after confirming the absence of contaminating cell types based on lineage marker expression. Six DC clusters were identified based on expression of canonical gene markers; four plasmacytoid clusters (pDC-1 – 4) due to high *Siglech*, *Ptprc*, and *Irf4* expression, two conventional dendritic cell 1-like clusters (cDC1–1 and 2) based on *Ccr7*, *Batf3*, *Id2*, and *Zbtb46* expression, and a single monocyte derived DC cluster (moDC) as determined by *Klf4*, *Cx3cr1*, and *Mrc1* expression (Fig 5B and S7 Table). All six clusters highly expressed *hFcRn* and *B2m* (Fig 5D).

To assess whether any dendritic cell populations responded to echovirus infection similarly to Kupffer cells, we examined changes in cell-type abundance across all eight conditions using bar plots. This analysis identified a single cluster, cDC1–1, that consistently expanded in response to infection across all four genotypes (Figs 5C and S4A). We confirmed this expansion by comparing the distribution of cDC1–1 cells between infected and uninfected samples using UMAP visualization (Fig 5E and S8 Table).

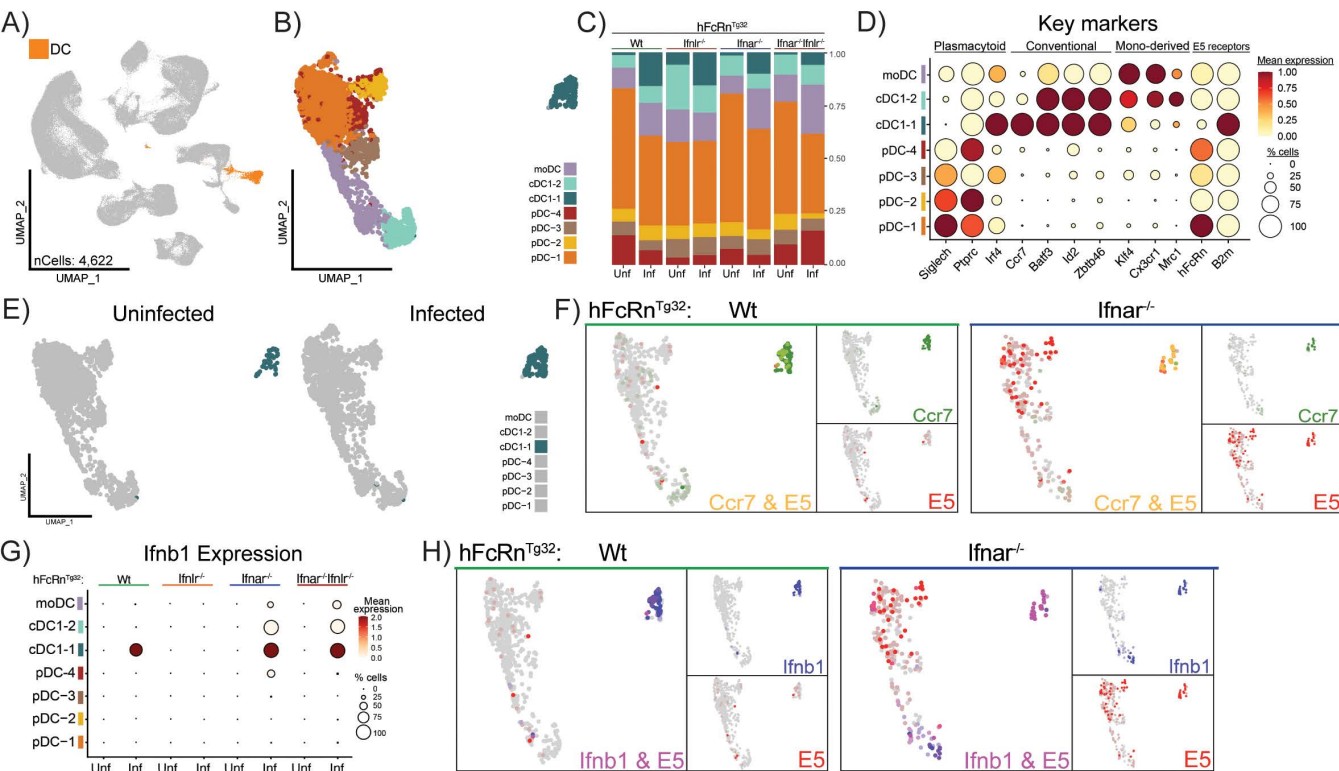

**Fig 5. A conventional dendritic cell 1–like population produces Type I IFNs in response to infection. A)** UMAP highlighting the dendritic cell (DC) cluster to be subsetted from the whole-liver dataset. This cluster is made up of 4,622 cells across all 8 experimental conditions. **B)** UMAP of the reclustered DC population, new cell identities were assigned based on key marker expression. Cell types are annotated as follows: plasmacytoid dendritic cells (pDC-1, orange), pDC-2, yellow), pDC-3- brown, pDC-4, red) conventional dendritic cells type 1-like (cDC1-1, dark blue, cDC1-2, light blue), and monocyte derived dendritic cells (moDC- purple). **C)** Bar plot showing the proportion of each cell cluster within the subsetted dataset, split by genotype (top) and infection status (bottom). **D)** Key markers (x-axis) for each cell type (y-axis) including E5 receptors are shown as a Dot plot. Scaled mean expression is represented by a color scale and percent of cell expressing each gene is demonstrated by the size of the dot. Labels on top of the graph indicate gross groupings of key genes. **E)** UMAP of the reclustered dendritic cell population split by infection status to highlight the difference in abundance of the cDC-1 cluster (dark blue). **F)** Merged Feature plots of infected Wt (left) and Ifnar1[-/-] (right) samples showing the expression of a key cDC1 marker *Ccr7* (green) and E5 (red). Yellow dots indicate that a given cell expresses both *Ccr7* and E5 (co-expression). **G)** Dot plot showing the average expression and proportion of cells expressing *Ifnb1* across the different cell types (y-axis) separated by genotype (top) and infection status (x-axis). **H)** Merged Feature plots of infected Wt (left) and Ifnar1[-/-] (right) samples showing the expression of *Ifnb1* (blue) and E5 (red). Purple indicates that a given cell expresses both *Ifnb1* and E5.

To explore whether this expanding cDC1–1 population was directly infected, we examined the localization of echovirus transcripts in infected samples. Using blended feature plots of echovirus (red) and the cDC marker *Ccr7* (green), we found clear co-localization in the cDC1–1 cluster across all genotypes (Figs 5F and S4E). However, viral transcripts were also detected in multiple other clusters, with broader distribution in Ifnar1[-/-] and Ifnar1[-/-]Ifnlr1[-/-] mice, suggesting that viral abundance and cell-type targeting are shaped by IFN signaling status (S4B, S4C Fig). These findings differ from earlier observations in immunocompetent mice, where echovirus RNA was detected almost exclusively in KCs. In contrast, low levels of viral RNA were observed within DCs, particularly the cDC1–1 subset, suggesting that these cells may acquire viral material in the setting of impaired Kupffer cell control.

To determine whether type I IFN production was induced by a single cluster or more ubiquitously throughout the DC dataset, we analyzed the mean *Ifnb1* expression across infected and uninfected genotypes. Like the Mono.mac dataset, *Ifnb1* expression in Wt pups mapped to a single cluster, cDC1–1 (Figs 5G and S4D). In Ifnar1[-/-] and Ifnar1[-/-]Ifnlr1[-/-] pups,

*Ifnb1* induction extended across a broader range of dendritic cell populations including cDC1–1, cDC1–2, pDC-4, and moDC, highlighting the failure of these genotypes to effectively respond to type I IFNs during infection. Finally, we examined whether cDC1–1 cells that harbored echovirus also expressed *Ifnb1*, as observed in Kupffer cells. Using FeaturePlots, we assessed co-localization of echovirus and *Ifnb1* transcripts and quantified double-positive cells. In Wt samples, 40% of echovirus-positive cDC1–1 cells also expressed *Ifnb1* (Figs 5H and S4F and S9 Table). Strikingly, in Ifnar1-/- and Ifnar1-/-Ifnlr1-/- samples, 100% of virus-positive cDC1–1 cells also expressed *Ifnb1*, indicating sustained or amplified cytokine production in the absence of downstream signaling. These findings underscore the role of cDC1–1 cells in both sensing echovirus and initiating type I IFN responses.

## Hepatocytes act as key responders to type I IFN signals during infection

Hepatocytes are highly susceptible to many viral infections [14,17,30,31,40–43], make up the bulk of liver mass (~80%), and serve as the liver's major metabolic cell type [44]. To better understand the role of hepatocytes in innate immune sensing and response during echovirus infection, we determined whether their ability to respond to type I IFNs is essential for host survival. We subset the hepatocyte cluster from the whole liver dataset and then mapped viral RNA expression across hepatocyte clusters (Fig 6A, 6B). Using dot plots we found that in Wt and Ifnlr1-/- mice, echovirus transcripts localized primarily to the Hepatocyte-3 cluster (~50% of cells) and to a lesser extent the Hepatocyte-1 cluster (~20%) (S5B Fig). In contrast, Ifnar1-/- and Ifnar1-/-Ifnlr1-/- livers showed widespread infection, with 80–100% of cells in all four hepatocyte clusters harboring virus (S5C Fig). We next determined whether infected hepatocytes were producing IFNs. In Wt and Ifnlr1-/- livers, expression of both type I and type III IFNs was nearly undetectable (Figs 6E and S5D, S5E). However, in Ifnar1-/- and Ifnar1-/-Ifnlr1-/- mice, we observed robust induction of *Ifnb1* and *Ifnl2* in the Hepatocyte-2 cluster, despite this cluster not harboring high levels of viral RNA. These findings suggest that in the absence of IFN receptor signaling, infected hepatocytes fail to control viral replication and instead attempt to compensate by producing their own type I and III IFNs. To confirm that these clusters represented *bona fide* hepatocytes, we examined expression of canonical hepatocyte markers (*Alb*, *Afp*, *Ttr*) and proliferation markers (*Top2a*, *Mki67*, *Rps12*) (Fig 6D and S10 Table). We also examined cluster proportions across genotypes and infection conditions. Consistent with the non-migratory nature of hepatocytes, the proportions of each cluster remained stable across all eight experimental conditions and genotypes (Figs 6C and S5A).

To test whether hepatocyte responsiveness to type I IFNs is required to control echovirus infection and promote survival, we generated a conditional knockout (cKO) mouse in which *Ifnar1* was selectively deleted in hepatocytes (Ifnar1fl/fl AlbCre) on the hFcRnTg32 background (S11 Table). Because Albumin-Cre activity is inefficient in neonatal mice [45], we conducted the following experiments in 6-week old adults. At this age, Cre-mediated recombination is robust enough to observe hepatocyte-specific recombination events. Hepatocyte-specific Ifnar1 knockout mice were challenged with a previously determined, sub-lethal, dose of echovirus (10⁴ PFU, IP) and compared to both negative and positive control genotypes: hFcRnTg32 Ifnar1fl/fl (no Cre activity; IFN-responsive liver) and hFcRnTg32 Ifnar1-/- (whole-body Ifnar knockout). Following infection, all Ifnar1-/- mice exhibited mortality by 3 dpi, while all hepatocyte-specific Ifnar1 cKO mice succumbed by 6 dpi (Fig 6F). In contrast, 60% of Ifnar1fl/fl control mice survived beyond 7 dpi, demonstrating that the absence of Ifnar1 specifically in hepatocytes is sufficient to cause mortality in an otherwise immunocompetent background. To confirm the specificity and consequences of the cKO, we quantified viral titers in key organs. In the liver, Ifnar1fl/fl AlbCre mice showed significantly higher viral loads than control mice (Figs 6G and S5F), confirming both the efficiency of hepatocyte-specific Ifnar1 deletion and the critical role of hepatocyte IFN signaling in viral restriction. In the brain, viral titers were markedly lower in cKO mice compared to whole-body Ifnar1-/- animals, indicating that the brain retained IFN responsiveness and could still mount a protective response. Pancreatic viral loads were comparable across all genotypes (S5F Fig), consistent with this organ being broadly permissive to echovirus infection regardless of IFN status. Blood viral titers (S5F Fig) confirmed systemic dissemination and excluded impaired access as the explanation for reduced brain infection. These

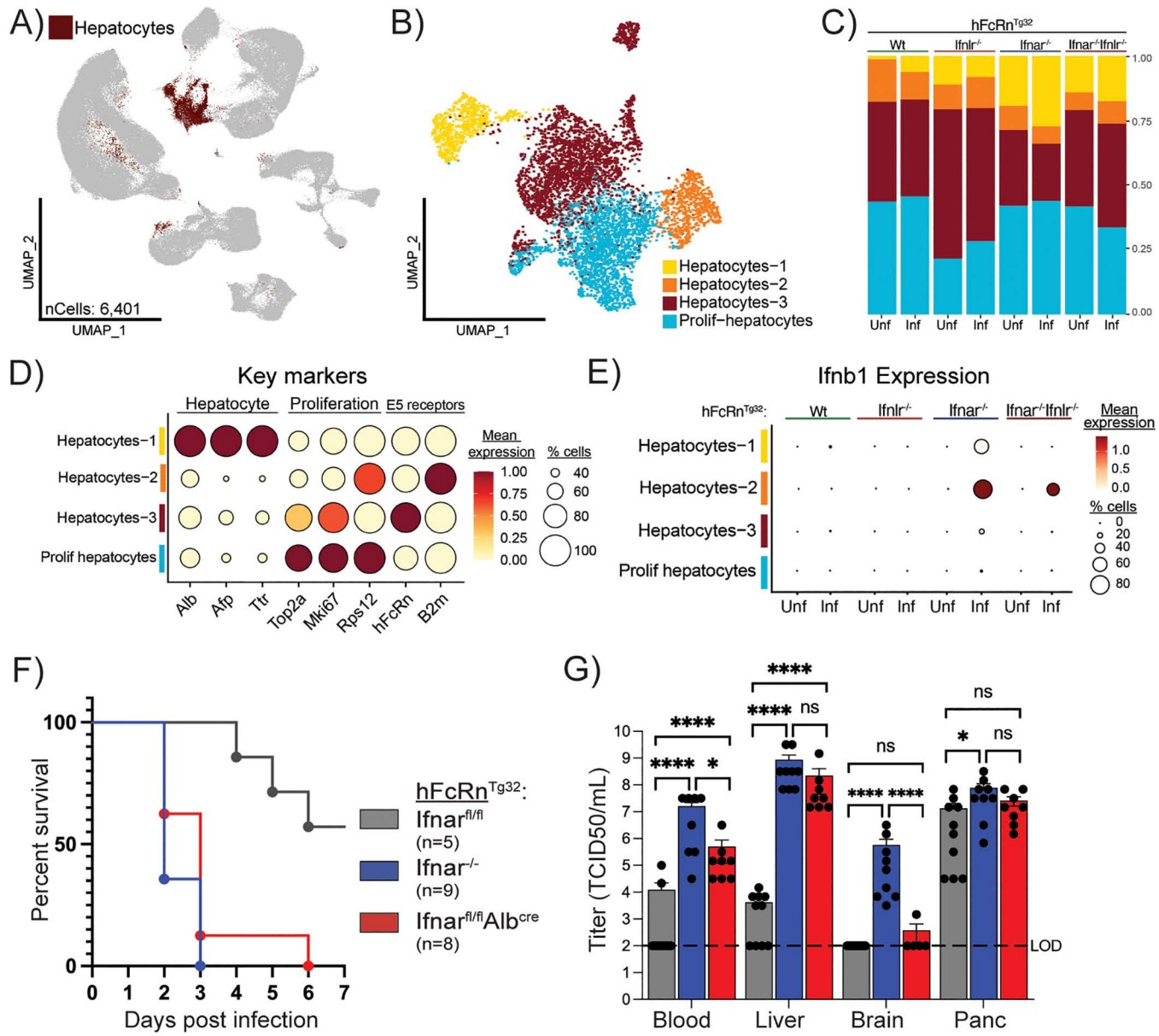

**Fig 6. Hepatocytes are major responders to type I interferon during infection. A)** UMAP highlighting the hepatocyte cluster to be subsetted from the whole-liver dataset. This cluster is made up of 6,401 cells across all 8 experimental conditions. **B)** UMAP of the reclustered hepatocyte population, new cell identities were assigned based on key marker expression. Cell types are annotated as follows: hepatocytes-1 (yellow), hepatocytes-2 (orange), hepatocytes-3 (brown), proliferating hepatocytes (prolif-hepatocytes, blue). **C)** Bar plot showing the proportion of each cell cluster contributes to the total cell composition of each experimental condition. Annotations include genotype (top), infection status (bottom), and total proportion (right). **D)** Dot plot of 6 key hepatocyte markers (x-axis) for each cluster (y-axis), including the two components of the E5 receptor complex. Scaled mean expression is represented by a color scale and the percent of cell expressing each gene is demonstrated by the size of the dot. Labels on top of the graph indicate gross groupings of key genes. **E)** Dot plot of the average expression and proportion of cells expressing *Ifnb1* across the different cell types (y-axis) separated by genotype (top) and infection status (x-axis). **F)** Survival curve of six-week-old control and experimental mouse genotypes infected with $10^4$ echovirus5, intraperitoneally. Animals that survived past one week post infection (1wpi) were euthanized. Genotypes are as follows: negative control hFcRcn^Tg32^Ifnar1^fl/fl^ (grey), positive control hFcRcn^Tg32^ Ifnar1^-/-^ (blue), and experimental cKO hFcRcn^Tg32^ Ifnar1^fl/fl^ Alb^Cre^ (red). **G)** Viral titers collected from key sites of infection (blood, liver, brain, pancreas) at 40hpi, the black dashed line indicated the limit of detection for this assay. Asterisks represent samples with statistically significant differences in viral titers across genotypes as determined by one-way ANOVA tests (* p < 0.0113,**** p < 0.0001, ns: not significant).

data demonstrate that hepatocyte-specific loss of *Ifnar1* is sufficient to recapitulate the mortality phenotype observed in whole-body knockouts following echovirus challenge. Our findings establish hepatocytes as critical recipients of protective type I IFN signals during infection.

## Discussion

Clinical morbidity and mortality during echovirus infection in neonates has long been attributed to acute liver failure [10,46–48]. However, the specific contribution of hepatocyte dysfunction to disease outcome has not been directly tested. In this study, we profiled the hepatic response to echovirus infection in neonatal mice across four mouse genotypes with distinct capacities for type I and III IFN signaling. Comparative analysis revealed that *Ifnar* expression is required in multiple liver-resident cell types to mount an effective antiviral response. We identified the primary hepatic sources of type I IFNs during early infection and demonstrated that *Ifnar* expression in hepatocytes alone is necessary for survival following echovirus challenge. These findings provide direct evidence that hepatocyte impairment is sufficient to drive mortality and position the liver as a central organ in determining echovirus disease outcome.

Importantly, this study was conducted in neonatal mice to reflect the age group most vulnerable to echovirus infection in humans. The neonatal immune system is inherently limited in both type I IFN production and responsiveness, and the liver itself is still undergoing developmental maturation, including the establishment of resident immune populations and functional architecture [25,49,50]. These factors likely contribute to the increased severity of echovirus disease in neonates, yet most prior studies of viral hepatitis or liver-targeted infections have used adult mouse models, which do not recapitulate this unique vulnerability. By modeling infection in neonates, we uncovered liver-specific innate immune dynamics that may be absent or attenuated in adults. In particular, the expansion and activation of Kupffer cells and cDC1s suggest that the neonatal liver relies heavily on tissue-resident immune cells to detect infection and initiate protective IFN responses. This model highlights the importance of developmental context in shaping antiviral immunity and underscores the critical need for age-appropriate model systems to develop targeted interventions for neonatal enteroviral diseases.

While immune responses at barrier surfaces are often dominated by type III IFNs to limit tissue damage [39], our findings show that this pathway is not essential for antiviral protection in hepatocytes during echovirus infection. Despite prior *in vitro* evidence showing robust type III IFN production by hepatocytes in response to viral infection [18,19,40], we found that the type III IFN pathway is dispensable for survival and viral clearance from the liver *in vivo*. Instead, our data support a model in which professional immune cells, including Kupffer cells, dendritic cells, and to some extent neutrophils, serve as the primary source of type I IFNs during early infection. In Ifnar-competent animals, this response is sufficient to rapidly contain viral replication in the liver and prevent disease progression. We propose that hepatocytes preferentially produce type III IFNs under homeostatic or low-inflammation conditions as a strategy to avoid the potentially damaging effects of type I IFNs on liver tissue. By relying on smaller populations of resident immune cells to initiate type I IFN responses, the liver may preserve tissue integrity while still mounting an effective antiviral defense. In support of this, we observed that hepatocyte *Ifnb1* and *Ifnl2* expression was only robustly induced in Ifnar1$^{-/-}$ and Ifnar1$^{-/-}$Ifnlr1$^{-/-}$ models, contexts in which the canonical IFN response is impaired and viral burden is high. This suggests that type III IFN production by hepatocytes may represent a compensatory 'last resort' response in the absence of immune cell-derived type I IFNs.

As the predominant cell type in the liver, hepatocytes are essential not only for metabolic function but also for maintaining organ integrity in the face of infection. Their ability to respond to immune signals is critical for containing viral replication and preserving tissue homeostasis [51]. Our findings using hepatocyte-specific Ifnar knockout mice demonstrate that loss of this signaling capacity renders the liver highly susceptible to echovirus infection and is sufficient to drive organismal death. These data support clinical observations that acute liver failure is a leading cause of echovirus-related mortality in neonates [52], and position hepatocyte dysfunction as a primary driver of disease severity. At the same time, our results also raise the possibility that impaired type I IFN responsiveness in non-parenchymal immune cells contributes to viral spread and systemic disease. If sentinel cells, such as Kupffer cells, dendritic cells, or neutrophils, cannot respond to type

I IFNs produced during initial infection, we propose that the host must rely on direct viral sensing to trigger downstream responses. This shift may delay the initiation of protective immunity, permitting broader viral dissemination before control mechanisms are fully engaged. Therefore, we suggest a model in which both hepatocyte susceptibility and the functional capacity of innate immune cells to initiate and respond to IFNs likely contribute to overall echovirus pathogenesis and this should be considered in future studies of enterovirus-induced organ failure.

A key finding of this study is the selective expansion of Kupffer cells during echovirus infection, accompanied by their dominant role in type I IFN production. While KCs are well known for their role in sensing and phagocytosing pathogens, expansion of this population is not a common feature across other viral infections. For example, LCMV and certain flaviviruses deplete or suppress KC populations as a means of immune evasion [53], and even in hepatitis B (HBV) and C (HCV) models, KC numbers tend to remain stable [54,55]. Expansion of the KC population in this model closely reflects a phenomenon observed during acute liver injury in which monocytes are recruited and reprogrammed to a KC transcriptional profile to account for the loss of embryonically derived KCs [56]. These cells were the primary producers of *Ifnb1* in the liver and maintained expression even in the absence of *Ifnar1,* indicating a feedback-independent mode of viral sensing and cytokine production. Their central role in IFN production was further supported by colocalization of *Ifnb1* and viral transcripts, although it remains possible that KCs are not productively infected but instead acquire viral RNA through phagocytosis. Regardless, the presence of viral RNA and robust *Ifnb1* expression suggests that Kupffer cells serve as autonomous sensors and early initiators of the hepatic antiviral program.

A smaller but distinct contribution came from a subset of conventional dendritic cells (cDC1–1), which also expanded during infection and demonstrated *Ifnb1* expression. Although colocalization with viral RNA suggests uptake or sensing, it may not reflect productive infection, as DCs are highly efficient at capturing viral material through endocytosis. In Ifnar-deficient models, cDC1–1 cells showed uniform *Ifnb1* expression, suggesting sustained or compensatory activation in the absence of downstream IFN signaling. These findings indicate that cDCs act in parallel with KCs to reinforce type I IFN production and may bridge the early innate response with the priming of adaptive immunity. Although liver-resident cDCs have not been previously described in the context of echovirus infection, they are known to participate in antiviral responses to other hepatotropic viruses. In HBV infection, cDCs contribute to antiviral defense primarily through antigen presentation, while plasmacytoid DCs are the dominant IFN producers [57]. In LCMV models, cDCs facilitate cross-presentation and T cell activation [58]. The ability of cDCs to express high levels of TLR3 and other pattern recognition receptors positions them to respond to viral RNA under certain inflammatory contexts. Our findings suggest that echovirus infection may uniquely activate liver-resident cDCs in an IFN-independent manner, enabling them to contribute to a locally contained antiviral response when systemic sensing is impaired.

Neutrophils constituted a third myeloid subset displaying a type I IFN-induction signature during echovirus infection. Like the monocyte/macrophage (Mono.mac) and DC clusters, neutrophils contained a discrete Ifnb1-expressing cluster that expanded upon infection and preferentially colocalized with E5 viral transcripts (S6 Fig). Unlike these other populations, however, neutrophils up-regulated a broad panel of antiviral ISGs even in Ifnar1-deficient mice (S6H Fig), a pattern consistent with the well-documented IFN-λ responsiveness of neutrophils at mucosal barriers [59]. The induction of *Ifnl2* by hepatocytes in the dataset points to a paracrine axis that may prime neutrophils when type I IFN feedback is compromised. These observations suggest that neutrophils complement other immune cells in the liver, Kupffer cells and cDC1s [15], in sustaining an intraparenchymal antiviral state, underscoring the coordinated contribution of multiple myeloid lineages to host clearance and survival during echovirus infection.

Although the hFcRn mouse system enables physiologically relevant echovirus infection and recapitulates key early immunopathologic features of human neonatal disease, there are limitations of the model. Several fundamental differences exist between murine and human liver biology. First, liver architecture differs in the number and organization of lobules, with rodents exhibiting greater lobulation [60]. Second, immune-parenchymal development follows distinct timelines: in humans, immune–stromal interactions begin earlier in liver ontogeny, influencing when and how innate immune cells

integrate into the forming lobular network [61]. Third, cellular composition and regional specialization vary, with human livers displaying greater zonation, leading to slower metabolic flux and nutrient gradient equilibration compared to murine models [62,63]. These differences may collectively influence IFN kinetics and the nature of cell–cell crosstalk observed in our experiments. Therefore, while the precise magnitude and hierarchy of Kupffer cell, dendritic cell, and hepatocyte IFN responses should be viewed as mechanistic rather than directly predictive of human outcomes, these models nonetheless offer a tractable, genetically resolvable system for dissecting antiviral circuits under conditions that cannot be studied in humans. Finally, it was necessary to perform the conditional *Albumin* knockout experiments in adult mice, as the *Albumin*-Cre system is only effective in adults and does not function in the neonatal context. As a result, these experiments cannot be directly extrapolated to neonatal physiology, which was the primary focus of this study. Nonetheless, our previous work demonstrated that type I IFNs restrict echovirus infection in adult mice, suggesting that similar antiviral mechanisms are likely operative in neonates [31]. While this represents an inherent limitation of the current model, the adult conditional knockout provides essential mechanistic insight into hepatocyte-intrinsic IFN responses that complement the neonatal infection studies presented here.

In summary, our study defines the cellular and molecular determinants of hepatic antiviral defense during echovirus infection in the neonatal liver. We show that Kupffer cells and a subset of conventional dendritic cells serve as the primary producers of type I IFNs in the liver, coordinating a localized innate immune response that protects surrounding parenchymal cells. Hepatocytes, while not major initiators of IFN production, rely critically on their ability to respond to these signals. Using a hepatocyte-specific Ifnar knockout model, we demonstrate that loss of IFN responsiveness in hepatocytes alone is sufficient to drive mortality, highlighting the liver's central role in determining disease outcome. These findings reveal a cooperative, multicellular antiviral network in which specialized myeloid cells initiate protective immunity that must be sensed by hepatocytes to prevent lethal liver failure, particularly in the vulnerable neonatal setting.

## Materials and methods

### Ethics statement

All animal procedures were performed in accordance with the recommendations in the National Institutes of Health Guide for the Care and Use of Laboratory Animals and were approved by the Duke University Institutional Animal Care and Use Committee (IACUC).

### Virus growth and purification

Echovirus 5 (Noyce) was purchased from the ATCC. Echovirus stocks were amplified in HeLa cells cultured in DMEM supplemented with 10% FBS. Cells were infected at ~80% confluency and incubated for 48 hours until cytopathic effect was observed in ~90% of the monolayer. Infected cultures were harvested and subjected to freeze-thaw lysis, followed by detergent treatment (10% Triton X-100) and clarification by ultracentrifugation (7200 RPM, 20 min, 16°C, SW28 rotor). Virus-containing supernatants were further purified by ultracentrifugation over a 30% sucrose cushion (27,000 RPM, 3 hr, 16°C) and resuspended in PBS. Final viral stocks were aliquoted and stored at –80°C.

### Virus quantification and sequencing

Viral titers were determined by standard plaque assay on HeLa cells. Briefly, 12-well plates were seeded with $1.8 \times 10^5$ cells per well and incubated overnight. The following day, 10-fold serial dilutions of viral stocks were prepared and 100 µL was added per well. After 1 hour of adsorption at room temperature on a plate shaker, cells were overlaid with 1% agarose in 2×MEM supplemented with 4% FBS, 2% penicillin/streptomycin, and 2% non-essential amino acids. Plates were incubated at 37°C for 48 hours, then fixed with 0.05% crystal violet in methanol following plug removal. Viral titers (PFU/mL) were calculated based on plaque counts and dilution factors. Viral genome identity was confirmed by unbiased

deep sequencing performed by Alex Greninger (University of Washington). Viral RNA was extracted from purified stocks, converted to cDNA, and subjected to metagenomic next-generation sequencing (NGS) using the Illumina platform. Reads were quality-filtered, aligned to the reference echovirus genome, and analyzed for sequence identity and contamination. Consensus sequences were >99.9% identical to the input strain, confirming purity and genomic integrity of the viral stocks.

## Genotype generation

To generate the hFcRnTg32 Ifnar1[-/-] Ifnlr1[-/-] line, we crossed previously established hFcRn[Tg32] Ifnar1[-/-] and hFcRn[Tg32] Ifnlr1[-/-] mice [31,55]. Offspring were genotyped using a custom Transnetyx panel (available upon request), and pups homozygous for both Ifnar1 and Ifnlr1 deletions and positive for the hFcRn[Tg32] transgene were selected and interbred until litters were uniformly homozygous. To generate hepatocyte-specific Ifnar1 knockout mice (hFcRn[Tg32] Ifnar1[fl/fl]Alb[Cre]), Ifnar1[fl/fl] mice (JAX #028256) and Albumin-Cre mice (JAX #003574) were each first crossed to hFcRn[Tg32] mice to generate hFcRn[Tg32] Ifnar1[fl/fl] and hFcRn[Tg32] Alb[Cre] lines. These lines were then intercrossed and backcrossed, three times, to obtain mice homozygous for hFcRn[Tg32], Ifnar1[fl/fl], and Alb[Cre]. Genotyping was performed at each generation using Transnetyx panels (available upon request). All animal procedures were approved by the Institutional Animal Care and Use Committee (IACUC) at Duke University and conducted in accordance with NIH guidelines for the care and use of laboratory animals.

## Mouse infections and virus quantification

Unless otherwise specified, the experimental unit was the individual mouse, and all analyses were performed at the level of the individual animal. Neonatal (P7) and adult (6-week-old) mice were infected intraperitoneally (i.p.) with 50 µL of echovirus diluted in PBS using 27G (neonates) or 25G (adults) needles. Based on prior work, inoculum doses were pre-specified and differed by genotype to capture early-acute survival windows: hFcRn[Tg32] Ifnar1[-/-] and hFcRn[Tg32] Ifnar1[-/-] Ifnlr1[-/-] neonates received 50 PFU, hFcRn[Tg32] and hFcRn[Tg32] Ifnlr1[-/-] neonates received $10^6$ PFU, and adult hFcRn[Tg32] Ifnar1[-/-], hFcRn[Tg32]Ifnar1[fl/fl], and hFcRn[Tg32]Ifnar1[fl/fl]Alb[Cre] received $10^4$ PFU. Following infection, neonates were transferred to surrogate dams to prevent maternal transmission. These surrogate dams were C57BL/6 females lacking the hFcRn transgene, which renders them resistant to echovirus infection and thus limits the risk of viral shedding or re-infection of pups. The number of mice per genotype ranged from 8 to 12 individuals, depending on litter size, as two separate litters were used for each experimental condition (survival and titer). Litters with fewer than four pups were excluded a priori. No animals were excluded from survival or titer analyses after enrollment. For single-cell RNA-sequencing (scRNA-seq), cell-level quality control criteria are described below. Within each genotype, pups were assigned to experiments using block randomization by litter to balance potential litter effects across experimental days and cages. Analyses using the mouse as the experimental unit accounted for litter as a random effect. Primary outcomes for model characterization included (i) survival to 7 days post-infection (dpi)—all Wt and Ifnlr[-/-] mice survived to this time point and were euthanized—and (ii) viral titers in blood, liver, pancreas, and brain at 40 hours post-infection (hpi). All experimental cages were monitored twice daily for signs of distress, and no unexpected adverse events were observed. Blinding was not feasible given the experimental design; however, primary endpoints were objective and not influenced by blinding status.

For survival analyses, animals were monitored daily and euthanized upon reaching humane endpoints (e.g., ≥25% weight loss or predetermined signs of morbidity). At euthanasia, tissues were collected for viral quantification. Blood was harvested into K$_2$EDTA microtainer tubes (BD, Ref #365974), and livers were collected into bead-beating tubes (Omni International, Cat #19–628; neonates) or gentleMACS M tubes containing 6 mL DMEM + 10% FBS (Miltenyi Biotec, Cat #130-093-236; adults). Brain and pancreas were collected in 4 mL DMEM in gentleMACS tubes and homogenized using the gentleMACS Dissociator (Programs RNA_02.01 or RNA_01.01). Tail snips were collected for genotyping (Transnetyx). All samples were snap-frozen and stored at –80°C until processing.

Frozen tissues were thawed at room temperature. Neonatal livers were homogenized in 500 µL DMEM + 10% FBS using a Bead Ruptor Elite (Omni International; 4 m/s, 20 s). After centrifugation (5 min, 12,000 rpm), 400 µL of supernatant was collected. Adult tissue homogenates were similarly prepared, and 1 mL aliquots were reserved for viral titering.

For viral titration of infected tissues, serial 10-fold dilutions were prepared in DMEM + 10% FBS. For blood, 1.2 µL was added to 119 µL media for the first dilution; for tissues, 12 µL homogenate was added to 108 µL media. Dilutions (100 µL) were transferred to 96-well plates pre-seeded with HeLa 7B cells at ~60% confluency and incubated at 37°C for 5 days. Plates were developed by replacing media with 0.05% crystal violet in 10% methanol for 15 minutes, rinsing briefly in water, and air-drying overnight.

For scRNA-seq experiments, the experimental unit was the individual liver, encompassing all cells isolated from that organ. All mice were euthanized at 40 hours post-infection (hpi), and livers were collected and processed for single-cell sequencing. Within each genotype, litters were randomly divided into two groups with an equal distribution of males and females: one group was designated for infection, and the other served as the uninfected control. Primary scRNA-seq outcomes included cell type composition, viral RNA detection and mapping, interferon and ISG induction scores, and transcriptional trajectory analyses.

## Luminex protein quantification

For protein quantification, whole-liver lysate samples previously collected for viral titer measurements were thawed and centrifuged at $10,000 \times g$ for 5 min to pellet residual cellular debris. Supernatants (100 µL per sample) were assayed according to the manufacturer's instructions using a custom IFN multiplex consisting of the Invitrogen Procarta-Plex Mouse IFN-α/IFN-β Panel 2-Plex (Cat. #EPX02A-22187-901) and the ProcartaPlex Mouse IL-28 Simplex (Cat. #EPX01A-26028-901). Protein concentrations (pg/mL) were used directly for subsequent analyses and figure generation.

## scRNA sequencing

Livers were harvested from uninfected and echovirus 5–infected neonatal mice (n = 3 per condition and genotype) at 40 hours post-infection. Each liver was dissected into ice-cold DPBS supplemented with calcium and magnesium, minced using sterile scalpels, and mechanically dissociated by syringe aspiration. Tissue suspensions were filtered through 100 µm strainers and centrifuged at $50 \times g$ for 5 minutes at 4°C to enrich hepatocytes. The resulting supernatant containing non-parenchymal cells (NPCs) was saved, and hepatocyte pellets were treated with red blood cell lysis buffer (Roche, Cat #11814389001) for 1 minute at room temperature. Lysis was quenched with ice-cold PBS + 1% FBS, and samples were passed through 70 µm filters. Hepatocyte and NPC fractions were combined, and viable cell counts were determined using trypan blue exclusion on an automated cell counter (Bio-Rad TC20). Up to $2 \times 10^7$ cells per sample were processed using the Dead Cell Removal Kit (Miltenyi Biotec, Cat #130-090-101) according to the manufacturer's instructions. Final cell suspensions were resuspended in PBS + 1% FBS and kept on ice prior to single-cell RNA-seq library preparation, which was performed at the Duke Sequencing and Genomic Technologies Shared Resource.

## Single-cell RNA-seq library preparation and data analysis

Libraries were prepared from 10,000 total cells per sample using the 10x Genomics Single Cell 3' Reagent Kit v2 (Manual Part #CG00052). Sequencing was performed on an Illumina NovaSeq 6000 (Illumina, San Diego), using an S2 flow cell for Wt and Ifnlr1[-/-] samples, and an S4 flow cell for Ifnar1[-/-] and Ifnar1[-/-] Ifnlr1[-/-] samples, achieving approximately 60,000 reads per cell. FASTQ files were processed using Cell Ranger v7.1.0 via the 10x Genomics Cloud Analysis pipeline, using a custom Mus musculus reference genome that included the Echovirus 5 genome and the human FcRn trans-gene (hFcRn). Filtered feature-barcode matrices were downloaded via command line and analyzed in R (v4.3.1) using a Seurat-based pipeline. The following packages were used: Seurat (v4.4.0), SeuratObject (v4.1.4), SeuratWrappers (v0.3.19), devtools (v2.4.5), dplyr (v1.1.3), harmony (v1.1.0), patchwork (v1.1.3), CellChat (v1.6.1), DESeq2 (v1.40.2),

EnhancedVolcano (v1.18.0), and ComplexHeatmap (v2.16.0). Filtered matrices were merged within each genotype, and cells were retained if they contained 500–7,500 detected genes, fewer than 75,000 total counts, and less than 10% mitochondrial transcripts. X- and Y-linked genes were annotated using Ensembl BioMart (mmusculus_gene_ensembl), and cells expressing red blood cell markers were excluded. SCTransform normalization was applied to each dataset, regressing out mitochondrial content, ribosomal content, gene/unique molecular identifier (UMI) counts, and sex-linked gene expression. Datasets were integrated using the SCTransform workflow (SelectIntegrationFeatures, PrepSCTIntegration, FindIntegrationAnchors, and IntegrateData), followed by PCA and UMAP dimensional reduction using the top 50 principal components and clustering with FindNeighbors and FindClusters. ALRA-based imputation (RunALRA) was performed prior to marker identification. All datasets were also normalized and scaled (NormalizeData, ScaleData) before Harmony-based integration to align cells by cell type rather than sample origin. Harmony embeddings were used for final dimensional reduction (PCA and UMAP, 40 PCs), and clustering was performed at resolution 0.2. Marker genes were identified using FindAllMarkers on ALRA-imputed data (minimum log fold change = 0.25, expressed in ≥25% of cells). For focused analysis of specific clusters, subsets were reprocessed with SCTransform and Harmony to resolve intracluster heterogeneity. Differential expression analyses across infection status were performed using DESeq2, and results were visualized using ComplexHeatmap. Pseudotime trajectories of infected cells were inferred using Slingshot, specifying a starting cluster and allowing the endpoint to be inferred. Additional analysis code is available on the CoyneLab GitHub repository (https://github.com/CoyneLabDuke/Echovirus-infected-neonatal-mouse-livers). All raw files have been deposited on SRA under the following BioProjectID: PRJNA1303135.

## Histology, hybridization chain reaction (HCR), and RNA scope imaging

Fresh liver lobes were collected and fixed overnight in 10% neutral buffered formalin (NBF), then rinsed and stored in 70% ethanol until processing. Fixed tissues were submitted to Histowiz for paraffin embedding, sectioning (horizontal orientation to maximize cross-sectional area), and hematoxylin and eosin (H&E) staining. Digitized H&E images were accessed via the Histowiz user portal.

For HCR, paraffin-embedded slides were deparaffinized through sequential xylene and ethanol washes, followed by heat-induced antigen retrieval in 1 × citrate buffer (pH 6.0) using a pressure cooker for 20 minutes. Slides were cooled to room temperature, rinsed in deionized water, and sample areas circled with a hydrophobic barrier pen. Sections were treated with 10 µg/mL proteinase K for 10 minutes at 37°C, then incubated with HCR hybridization buffer (Molecular Instruments) for 10 minutes at 37°C. Custom probes targeting Echovirus 5 RNA (previously described [64]) were applied in hybridization buffer and incubated overnight at 37°C in a humidified chamber. The following day, slides were washed in probe wash buffer and incubated in 5 × SSCT at room temperature for 3 hours. HCR amplification was initiated by incubating slides in amplification buffer for 30 minutes. Snap-cooled fluorescent HCR hairpins were prepared by heating to 95°C for 90 seconds, then allowed to equilibrate at room temperature in the dark for 30 minutes. Hairpins diluted in amplification buffer were applied to the slides and incubated overnight at room temperature. On Day 3, slides were washed in 5 × SSCT, counterstained with DAPI in PBS for 5 minutes, and mounted using Vectashield (Vector Laboratories, Cat #H-1200–01). Coverslips were sealed with nail polish, and slides were imaged using an Olympus FV3000 confocal laser scanning microscope and images were prepared using FIJI.

For RNAScope. Day 1, samples were treated as previously described [65]. Formalin-fixed paraffin-embedded (FFPE) liver sections were baked at 60°C for 1 hr, deparaffinized, and allowed to dry for 5 min at 60°C. Hydrogen peroxide was applied for 10 min at room temperature (RT), followed by a brief water rinse. Slides were then placed in target retrieval buffer and incubated at >99°C for 15 min (InstantPot method). After a brief water rinse, slides were immersed in 100% ethanol for 3 min at RT, dried for 5 min at 60°C, and hydrophobic barriers were drawn around the tissue sections. Protease Plus was added, and slides were incubated at 40°C for 30 min. During this step, probe solutions were prepared by diluting Channel 2 (C2) or Channel 3 (C3) probes 1:1 in the Channel 1 (C1) probe solution or probe diluent. Probes were

prewarmed at 40°C for 10 min and cooled to RT for 20 min before use. Slides were washed and hybridized with the diluted probe solution for 2 hr at 40°C, washed in wash buffer, and stored overnight in 5 × SSC buffer. On day 2, slides were sequentially incubated at 40°C with AMP1, AMP2, and AMP3 reagents for 30 min, 30 min, and 15 min, respectively, with washes between each step. HRP reagent was then applied for 15 min at 40°C, followed by incubation with a diluted fluorophore (TSA Vivid 520 or 650, 1:1500 in TSA buffer) for 30 min at 40°C. Slides were washed, treated with HRP blocker for 15 min at 40°C, and this process was repeated for each detection channel with a distinct fluorophore. After hybridization and detection, slides were incubated with an autofluorescence quenching reagent (Vector Labs, Cat. #SP-8400) for 5 min at RT, counterstained with DAPI for 30 sec, and mounted in Vectashield antifade medium (Cat. #H-1200–01). Images were acquired using an Olympus FV3000 confocal laser scanning microscope and processed with FIJI software. A detailed version of the RNAscope protocol is available from ACDbio.

## Statistics and reproducibility

All experiments were independently reproduced using tissue samples from multiple litters. Statistical analyses were performed using Prism (GraphPad Software), and data are presented as mean ± standard deviation (SD) unless otherwise noted. Statistical tests used for each experiment are detailed in the corresponding figure legends. One-way ANOVA was used to assess differences in viral titers across multiple genotypes. A p value of <0.05 was considered statistically significant, with exact p values provided in the figure legends.

## Supporting information

**S1 Fig. Type I IFN deficiency increases viral burden and mortality during echovirus infection. A)** Viral titers of infected neonatal mice split by genotype and sex. Striped bars indicate female mice and solid bars represent male mice. Grey bars reflect blood samples and red bars reflect liver samples. Limit of detection for the assay is marked by a dashed black line. There was no significant difference (as determined by one-way ANOVA tests) between titers taken from male or female mice within each genotype across blood and liver. **B)** H&E images (scale bar, 50mm) of uninfected, age-matched, livers across all four genotypes. Infiltrating immune cells are indicated by yellow arrows.
(TIF)

**S2 Fig. Sex does not influence E5 viral burden or liver cellular composition during early acute infection. A)** Dot plots visulaizing expression of X and Y linked genes for each mouse, groupe by genotype (top) and experiemntal condition (bottom). Female and male mice were equally represented across the dataset. **B)** Cellular composition bar plots for each mouse (left) representing Wt and Ifnlr1$^{-/-}$ genotypes (top), inclding the number of cells sequenced, nCells (right). **C)** Dot plot of scaled E5 transcript in the liver across Wt and Ifnlr1$^{-/-}$ genotypes split by individual mouse. Mean expression and percent of cells in which E5 transcript is found is represented by color and dot size. **D)** Cellular composition Bar plots and number of cells sequenced (nCells) split by individual mouse for Ifnar1$^{-/-}$ and Ifnar1$^{-/-}$ Ifnlr1$^{-/-}$ genotypes. **E)** Dot plot of scaled E5 transcript in the liver across Ifnar1$^{-/-}$ and Ifnar1$^{-/-}$ Ifnlr1$^{-/-}$ genotypes split by individual mouse Mean expression and percent of cells in which E5 transcript is found is represented by color and dot size.
(TIF)

**S3 Fig. Kupffer cell dynamics during echovirus infection are consistent between mice within each genotype. A)** Bar plots depicting the proportions each cluster contributes to the overall composition of the condition split by mouse (left) and genotype (top) with nCells noted for each sample (right). **B)** Slingshot trajectories superimposed and labeled (Prolif, Inflam, or KC) on the reclustered Mono.mac UMAP (left panel). Feature plot of the pseutotime values (0–100) assigned to each cell along the proliferation trajectory (right panel). Color gradient (blue to red) and arrow indicate the directionality of cell differentiation along this pathway. **C)** Dot plot of *Ifnb1* induction by cell type (y-axis) split by genotype (top) and infection status

(x-axis) for all mice present in the dataset. Expression data was scaled to allow for comparison of *Ifnb1* response across genotypes with color reflecting mean gene expression and dot size reflecting proportion of the cells in each cluster that contain the gene of interest. **D)** Dot plots of scaled mean expression of E5 split by cell type (y-axis), genotype (top), and mouse (x-axis). Left panel reflects Wt and Ifnlr1<sup>-/-</sup> average proportion of E5 containing cells for each cluster (between 0–50%). In contrast, the right panel reflects Ifnar1<sup>-/-</sup> and Ifnar1<sup>-/-</sup> Ifnlr1<sup>-/-</sup> genotypes whose average proportion of E5 containing cells for each cluster is between 25–100%. **E)** Merged Feature plots looking at expression of key KC marker, *Marco* (green) and E5 (red) across infected Ifnlr1<sup>-/-</sup> and Ifnar1<sup>-/-</sup> Ifnlr1<sup>-/-</sup> samples. Cells expressing both genes appear as yellow dots. **F)** Merged Feature plot showing expression of *Ifnb1* (blue) and E5 (red) in infected Ifnar1<sup>-/-</sup> Ifnlr1<sup>-/-</sup> samples. Cells expressing both genes appear as purple dots. The Ifnlr1<sup>-/-</sup> samples did not induce detectable *Ifnb1* in response to infection, therefore, a complimentary Feature plot could not be produced for this genotype.
(TIF)

**S4 Fig. Dendritic-cell infection dynamics are consistent regardless of sex for each genotype. A)** Bar plots depicting the proportions each cluster contributes to the overall composition of the experimental condition. Plots are split by genotype (top), individual mouse (left), and proportion (bottom), with nCells annotated to the right of each sample. **B)** Dot plot of *Ifnb1* production by cell type, split by genotype and infection status for all mice present in the dataset. Expression data was scaled to allow for comparison of *Ifnb1* response across genotypes with larger circles representing a greater proportion of the cells in that cluster having reads and darker red indicating clusters with higher gene expression. **C)** Dot plots of scaled mean expression of echovirus split by cell type (y-axis), genotype (top), and mouse (x-axis) for Wt and Ifnlr1<sup>-/-</sup> samples. **D)** Dot plots of scaled mean expression of echovirus split by cell type (y-axis), genotype (top), and mouse (x-axis) for Ifnar1<sup>-/-</sup> and Ifnar1<sup>-/-</sup> Ifnlr1<sup>-/-</sup> samples. **E)** Merged Feature plots looking at co-expression of key dendritic cell marker, *Ccr7* in green, and E5 in red across infected Ifnlr1<sup>-/-</sup> (left) and Ifnar1<sup>-/-</sup> Ifnlr1<sup>-/-</sup> (right) samples. Cells expressing both genes appear as yellow dots. **F)** Merged Feature plot showing co-expression of key *Ifnb1* in blue and E5 in red in infected Ifnar1<sup>-/-</sup> Ifnlr1<sup>-/-</sup> samples. Cells expressing both genes appear as purple dots. Ifnlr1<sup>-/-</sup> samples did not induce detectable *Ifnb1* in response to infection, therefore, a complimentary Feature plot could not be generated for this genotype.
(TIF)

**S5 Fig. Echovirus burden in the liver and other key secondary sites of infection is not affected by sex. A)** Cellular proportion bar plots and number of cells sequenced (nCells) split by individual mouse for all genotypes. Annotations include genotype (top), infection status (left), total proportion (bottom), and nCells (right). **B)** Dot plot showing the average expression and proportion of cells expressing echovirus-5 across the different hepatocyte clusters (y-axis) separated by genotype (top) and individual mouse (x-axis). **C)** Dot plot of scaled E5 transcript in hepatocytes across Wt and Ifnlr1<sup>-/-</sup> genotypes split by individual mouse (x-axis) and grouped by genotype (top). **D)** Dot plot showing scaled *Ifnb1* transcript across hepatocytes clusters and genotypes split by individual mouse (x-axis) and grouped by genotype (top). **E)** Dot plot of scaled *Ifnl2* transcript across hepatocytes clusters and genotypes split by individual mouse (x-axis) and grouped by genotype (top). **F)** Echovirus titers from key secondary sites of infection including the blood, liver brain, and pancrease. Titers were determined by TCID50 on organs harvested from all three genotypes (hFcRn<sup>Tg32</sup>: Ifnar1<sup>-/-</sup>, Ifnar1<sup>fl/fl</sup>, Ifnar1<sup>fl/fl</sup> Alb<sup>Cre</sup>), split by sex to confirm the lack of sex-bias during infection. Limit of detection for the assay is marked by a dashed black line. There was no significant difference (as determined by one-way ANOVA tests) between titers taken from male and female mice within each genotype across all four organs.
(TIF)

**S6 Fig. Cd14<sup>Hi</sup> neutrophils produce type I IFNs in response to infection. A)** UMAP highlighting the three neutrophil clusters in the original whole-liver dataset. These populations consist of 87,861 cells across all 8 experimental conditions. **B)** UMAP of the reclustered neutrophil population with the original identities assigned: Neutrophil-1 (light pink),

Neutrophil-2 (magenta), Neutrophil-3 (purple). **C)** UMAP of the reclustered neutrophil population with new cell identities were assigned based on key marker expression. Cell types are annotated as follows: Ly6g$^{Hi}$ (-1, dark pink, and -2, light pink), Mmp9$^{Hi}$Cd55$^{Hi}$ (-1, dark purple and -2, light purple), Erythrocyte-like (brown), Cd14$^{Mid}$ (light blue), Cd14$^{Hi}$ (dark blue), Mmp9$^{Hi}$Cd55$^{Mid}$ (Green), Megakaryocyte-like (orange), mitochondrial-high (mt$^{Hi}$, grey), and Top2a$^{Hi}$Mki67$^{Hi}$ (yellow). **D)** Bar plot showing the proportion of each cell cluster within the subsetted dataset, split by genotype (top) and infection status (bottom). **E)** Key markers (genes on x-axis) for each cell type (y-axis) including E5 receptors are shown as a Dot plot. Scaled mean expression is shown as a color scale and percent of cell expressing each gene is reflected by the size of the dot. Labels on top of the graph indicate gross groupings of key genes. **F)** UMAP of the reclustered dendritic cell population split by infection status to highlight the difference in abundance of the Cd14$^{Hi}$ cluster (dark blue). **G)** Dot plot showing the average expression and proportion of cells expressing *Ifnb1* across the different cell types (y-axis) separated by genotype (top) and infection status (x-axis). **H)** Dot plot showing the average expression and proportion of cells expressing panel of genes used for the ISG-score across genotype and infection status. **I)** Merged Feature plots of infected Wt (left) and Ifnar1$^{-/-}$ (right) samples showing the co-expression of a key Cd14$^{Hi}$ marker *Cd14* (green) and E5 (red). Yellow indicates that a given cell expresses both *Cd14* and E5. **J)** Merged Feature plots of infected Wt (left) and Ifnar1$^{-/-}$ (right) samples showing the co-expression of *Ifnb1* (blue) and E5 (red). Purple indicates that a given cell expresses both *Ifnb1* and E5.
(TIF)

**S1 Table. Enriched marker genes identified in neonatal mouse liver scRNA-Seq.** Marker genes enriched in liver cell populations identified by single-cell RNA sequencing of whole livers across four different IFN knockout mouse genotypes. Markers were determined using differential expression analysis with a log2 fold change threshold of log2FC > 0.25 and adjusted p-value adj p < 0.05.
(CSV)

**S2 Table. Normalized read counts (DESeq2) from all experimental conditions.** Normalized gene count matrix generated by DESeq2 package "NormCounts". Data normalized across all eight conditions for each liver sample.
(TXT)

**S3 Table. Differential gene expression table for ISG score generation.** List of differentially expressed genes between Wt-infected livers and Wt-uninfected livers with a log$_2$fold change greater than 4. List of differentially expressed genes between Wt-infected livers and Ifnar$^{-/-}$Ifnlr$^{-/-}$ infected livers with a log2fold change greater than 4. List of overlapping genes from the two, previously described lists.
(XLSX)

**S4 Table. Enriched marker genes identified from the Mono.Mac subclustered population.** Marker genes enriched in the Mono.mac subclustered data across all eight datasets. Markers were determined using differential expression analysis with a log2 fold change threshold of log2FC > 0.25 and adjusted p-value adj p < 0.05.
(CSV)

**S5 Table. Number of cells in the Mono.Mac subcluster split by cell type and infection status.** Number of cells present in each cell cluster in the Mono.mac subclustered dataset, split by infection status.
(XLSX)

**S6 Table. Mono.mac subcluster number of cells in which E5 and Ifnb1 map, split by cell type and genotype.** Number of cells that positively map *Ifnb1* or echovirus-5 reads and number of cells that both *Ifnb1* and echovirus-5 map in the Mono.mac subcluster, split by genotype.
(XLSX)

**S7 Table. Enriched marker genes identified from the DC subclustered population.** Marker genes enriched in the DC subclustered data across all eight datasets. Markers were determined using differential expression analysis with a log2 fold change threshold of log2FC > 0.25 and adjusted p-value adj p < 0.05.
(CSV)

**S8 Table. Number of cells in the DC subcluster, split by cell type and infection status.** Number of cells present in each cell cluster in the DC subclustered dataset, split by infection status.
(XLSX)

**S9 Table. DC subcluster number of cells in which E5 and Ifnb1 map, split by cell type and genotype.** Number of cells that positively map *Ifnb1* or echovirus-5 reads and number of cells that both *Ifnb1* and echovirus-5 map in the DC subcluster, split by genotype.
(XLSX)

**S10 Table. Enriched marker genes identified from the hepatocyte subcluster population.** Marker genes enriched in the hepatocyte subclustered data across all eight datasets. Markers were determined using differential expression analysis with a log2 fold change threshold of log2FC > 0.25 and adjusted p-value adj p < 0.05.
(CSV)

**S11 Table. Transnetyx genotype panels for conditional knockout mice and controls.** Genotyping panels used in Transnetyx to confirm the generation of the conditional knockout mouse (hFcRn$^{Tg32}$ Ifnar$^{fl/fl}$ Alb$^{Cre}$), and two control genotypes (hFcRn$^{Tg32}$ Ifnar$^{fl/fl}$ & hFcRn$^{Tg32}$ Ifnar$^{-/-}$).
(XLSX)

## Acknowledgments

We would like to acknowledge the assistance of the Molecular Genomics Core at the Duke Molecular Physiology Institute, Duke University School of Medicine, for the generation of data for the manuscript. We thank Megan Baldridge (Washington University) for providing Ifnlr1$^{-/-}$ mice, Sujan Shresta (La Jolla Institute for Immunology) for providing hFcRn$^{Tg32}$-Ifnar$^{-/-}$ mice, and Alexandra Wells (UT Southwestern) for assistance with the generation of hFcRn mouse strains. This project was supported by NIH R01-AI150151 (C.B.C). The funders had no role in study design, data collection and analysis, decision to publish, or preparation of the manuscript.

## Author contributions

**Conceptualization:** Emma Heckenberg, Carolyn B. Coyne.

**Data curation:** Emma Heckenberg.

**Formal analysis:** Emma Heckenberg, Carolyn B. Coyne.

**Funding acquisition:** Carolyn B. Coyne.

**Investigation:** Emma Heckenberg, Jacob G. Davis, Caitlin Hale.

**Methodology:** Emma Heckenberg.

**Project administration:** Carolyn B. Coyne.

**Supervision:** Carolyn B. Coyne.

**Visualization:** Emma Heckenberg, Carolyn B. Coyne.

**Writing – original draft:** Emma Heckenberg, Carolyn B. Coyne.

**Writing – review & editing:** Emma Heckenberg, Jacob G. Davis, Caitlin Hale, Carolyn B. Coyne.

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
