## [Decision Letter · Decision Letter 0]

10 Sep 2025

Tissue-Resident Macrophage and Dendritic Cell Cooperation Drives Type I IFN Immunity to Enteroviruses in the Liver

PLOS Pathogens

Dear Dr. Coyne,

Thank you for submitting your manuscript to PLOS Pathogens. After careful consideration, we feel that it has merit but does not fully meet PLOS Pathogens's publication criteria as it currently stands. Therefore, we invite you to submit a revised version of the manuscript that addresses the points raised during the review process.

Reviewers all agree that your study addresses an important question with robust scRNA-seq data. However, all highlight that the conclusions are overstated given the current evidence. In particular, additional validation beyond scRNA-seq is essential, such as interferon protein measurements, immunofluorescence or flow cytometry for viral RNA/protein co-localization, and pathology scoring/quantification are required to substantiate key claims. The attribution of IFN-I sources to Kupffer cells, cDCs, or monocytic macrophages also needs direct experimental support or should be toned down. Finally, the translational implications should be moderated, with clearer discussion of the limitations of the mouse model and use of a single viral strain.

Please submit your revised manuscript within 60 days Nov 09 2025 11:59PM. If you will need more time than this to complete your revisions, please reply to this message or contact the journal office at plospathogens@plos.org. Please include the following items when submitting your revised manuscript:

We look forward to receiving your revised manuscript.

Kind regards,

Helene Minyi Liu, Ph.D.

Academic Editor

PLOS Pathogens

Michael Letko

Section Editor

PLOS Pathogens

Editor-in-Chief

PLOS Pathogens

orcid.org/0000-0003-2946-9497

Editor-in-Chief

PLOS Pathogens

orcid.org/0000-0002-7699-2064

**Journal Requirements:**

At this stage, the following Authors/Authors require contributions: Emma Heckenberg, Jacob Davis, Caitlin Hale, and Carolyn B Coyne. Please ensure that the full contributions of each author are acknowledged in the "Add/Edit/Remove Authors" section of our submission form.

https://journals.plos.org/plospathogens/s/submission-guidelines#loc-parts-of-a-submission

5) We notice that your supplementary Figures, and Tables are included in the manuscript file. Please remove them and upload them with the file type 'Supporting Information'. Please ensure that each Supporting Information file has a legend listed in the manuscript after the references list.

6) Please ensure that the funders and grant numbers match between the Financial Disclosure field and the Funding Information tab in your submission form. Note that the funders must be provided in the same order in both places as well.

**Reviewers' Comments:**

Reviewer's Responses to Questions

**Part I - Summary**

Reviewer #1: In this work, Heckenberg and colleagues study the cell tropism and innate immune response to echovirus 5 infection using a neonatal transgenic mouse model. Echoviruses are important paediatric pathogens with a significant disease burden in neonatal populations. The liver is one of the secondary targets for echovirus infection, and the role of different liver cells during infection is poorly characterised. To better understand the role of each cell type and to focus on neonatal infections, the authors use transgenic neonatal mice expressing the human neonatal fragment crystallizable receptor (FcRn), which has been previously identified as a pan-echovirus receptor. Further, four genotypes of this transgenic mouse model, viz., immunocompetent (ifnar1+/ifnlr1+), ifnar1- (deficient in type 1 signalling), ifnlr1- (deficient in type 3 signalling), and ifnar1-/ifnlr1- double knockout (deficient in type 1 and 3 signalling), are used to better understand the innate immune responses following echovirus 5 infection. The study presented is primarily an in-depth analysis of single-cell RNA sequencing (scRNAseq) of the liver tissue from these neonatal mice. Using this scRNAseq data, the authors identify Kupffer cells and conventional dendritic cells (cDCs) as the main cells containing viral RNA in wild-type and type 3-deficient mice. Further, viral tropism is shown to expand to hepatocytes in both genotypes deficient in type 1 signalling. They then go on to show that Kupffer cells and cDCs are the primary producers of type 1 IFN. This leads them to conclude that the Kupffer cells are the primary source of type 1 IFN, and are supported by cDCs in driving this response. The authors also suggest that the role of type 3 signalling is limited, even though this has been identified as critical in vitro. Finally, using a conditional knockout of ifnar1 in the hepatocytes of adult mice, it is shown that hepatocytes are critical responders to type 1 signalling. These results position Kupffer cells and cDCs as central to antiviral defence in the neonatal mouse liver following E5 infection.

Novelty:

- Identification of the role of different cell types, and in particular, Kupffer cells and cDCs during infection

- Use of the neonatal mouse model, rather than the commonly used adult mouse model

Strengths:

- scRNA seq to probe the role of the various cell types during infection

- Use of different mouse genotypes to comprehensively evaluate the role of type 1 and type 3 signalling

Weakness:

- Reliant solely on scRNA seq data without any further validation or measurement of protein levels

- Limited translation potential, as the value of the mouse data to human infections is unclear

- Study limitations, including translational relevance and model system constraints, are not sufficiently addressed in the manuscript text

- A single Echovirus serotype, Echovirus 5, and a single strain (Noyce) of this serotype are used to generalise findings for echovirus infections

- Noyce strain has been shown to interact with cell surface heparan sulfate, which can be different between mice and humans

- Use of adult mice for conditional hepatocyte KO is a technical necessity, but this is not discussed and assumed to be similar in the neonatal setting

- The viral titres used for the genotypes are significantly different, and while the reasons are clear, it can have an impact on the cellular response. This needs to be discussed further.

Overall, this manuscript expands our knowledge on a clinically relevant pathogen and includes a robust experimental design as well as the use of a state-of-the-art technique. However, the general conclusions of the study are not warranted in light of the limitations highlighted.

Reviewer #2: In this study, the authors investigate the roles of type I (IFN-I) and type III (IFN-III) interferons in echovirus infection using hFcRn(Tg32) mice, which permit viral replication. They further employed scRNA-seq to identify cells producing and responding to IFNs. Their findings demonstrate that IFN-I, but not IFN-III, is critical for protection against echovirus in this model. Type I IFN signaling in hepatocytes conferred host survival, while IFN-β appeared to be produced mainly by Kupffer cells and cDC1-1 in the liver.

Reviewer #3: Heckenberg et al. utilize a combination of knockout mice and single-cell sequencing to examine the role of type I and III interferon pathways in the development of Echovirus 5 (E5)-induced liver disease in neonates. They identify Kupffer cells and hepatocytes as the cells most heavily infected by E5. They discover that Kupffer cells and a subset of dendritic cells within the liver are the main sources of type I interferon. Additionally, they demonstrate that mice lacking the type III interferon receptor can mount a protective response to E5 infection. In contrast, mice lacking the type I interferon receptor die from fatal disease after intraperitoneal (IP) inoculation. Because of the importance of the type I IFN response, the authors further show that hepatocytes need type I IFN receptors to prevent fatal infection.

The paper is well-written with clearly described and logical experimental approaches. For the most part, I was convinced by the data. The paper addresses an important aspect of Echovirus pathogenesis and will be of general interest to virologists. However, see critiques below.

1) I had several concerns regarding the interpretation of data from the Ifnar-/-Ifnlr-/- mice. In Figure 1, the authors show that viral titers are not significantly different from WT mice in the liver of the Ifnlr-/- mice. However, the level of inflammatory infiltrates was not quantified in Fig. 1C. It seems possible that differences in inflammation levels may exist between groups despite similar viral titers. It would be helpful to see pathology scoring performed for the H&E histology. Comparing the images shown for the Ifnar-/- and the Ifnar-/-Ifnlr-/- mice, there appears to be more inflammatory infiltrate in the DKO mice.

2) Related to the comment above, there seem to be more cells containing viral RNA in the Ifnar-/-Ifnlr-/- DKO than in the Ifnar-/- mice. Providing some quantification of the HCR data would be helpful here, or RT-qPCR data to measure viral positive and negative sense RNA in the different groups. In particular, examining negative-sense viral RNA would strengthen the analysis, as this is likely not detected in the scRNA-seq data.

**Part II – Major Issues: Key Experiments Required for Acceptance**

Reviewer #1: - Please quantify the interferon protein levels in the different genotypes, at least in the bulk liver lysates

- Immunofluorescence for localisation of specific proteins, as well as dsRNA or viral protein, with cell markers, is necessary to support the conclusions

- The data presented in Figure 3 is based on the intersection between two gene sets, viz., the wild type vs type 1 deficient. Based on this, 17 shared ISGs are identified. However, without a similar analysis for type 3, the role of type 3 during infection cannot be precluded. Please perform this analysis.

Reviewer #2: The data confirm the indispensable role of IFN-I; however, the conclusions regarding its cellular sources rely solely on scRNA-seq, without complementary experimental validation. The contribution of Kupffer cells and cDC1s to IFN-I–mediated restriction was not directly tested. Monocytic macrophages and DCs also expressed ISGs (Fig. 2), suggesting additional roles. Depletion experiments targeting Kupffer cells or cDCs in vivo, or CD11c-cre/Clec4f-cre × Ifnar flox models, would clarify their specific contributions in IFN production and responses.

In addition, viral RNA in scRNA-seq is used as the only evidence of infection. As the authors acknowledge, viral RNA in phagocytes could reflect uptake of infected material. Confirmation by staining viral proteins/RNAs in situ (as in Fig. 1D) with co-localization of the proposed infected cells via confocal microscopy or flow cytometry would strengthen this conclusion.

Fig. 1D shows more disseminated viral RNA in Ifnar/Ifnlr double KO mice compared with Ifnar KO, but this is not reflected in TCID50 data (Fig. 1B). Quantification of imaging would help clarify a potential role of IFN-III in the absence of IFN-I.

Reviewer #3: 1) Scoring of H&E pathology from the different groups.

2) Either quantification of the HCR data or perhaps more easily qRT-PCR to quantify positive and negative-sense viral RNA from the different groups.

**Part III – Minor Issues: Editorial and Data Presentation Modifications**

Reviewer #1: - Line 58, it is stated that fatality rises to 78%. Please specify that this is the case fatality ratio.

- In line 119, it is stated that wt and ifnlr1- mice survived beyond this early acute time point and refers to figure 1A. How long was this monitored, and what was the percentage survival?

- In line 156, 40 hpi is chosen for scRNA seq. What is this based on? Please provide a rationale for this choice in this section.

- In lines 169-176, markers for the various cells are listed. Where do these markers come from? The dataset or libraries used for the clustering are also not clear. Please add this to the methods.

- Line 359 says cDC cooperation with Kupffer cells. The data presented is not sufficient to claim this and only shows that cDCs also have a type 1 response. Please adjust.

- Line 548 of the discussion states that it remains possible that KCs are not productively infected. Yet, the results section is presented as if they are. This needs to be stated earlier, and the results adjusted accordingly. As these cells as phagocytic, this cannot be dismissed without evidence.

- Based on the scRNA seq analysis presented from line 671 onwards, it appears that doublets and multiplets have not been removed. Please explain why?

- The QC for the scRNA seq is not described sufficiently, and the QC data are not presented. Please add this.

- It is highly recommended that the ARRIVE guidelines be followed for the reporting of the in vivo experiments.

- Ethical statements for the animal work are missing.

In addition to these, the broad conclusions of the mouse work on human infection must be tempered. Furthermore, the limitations of the study must be discussed in detail. Please include the following:

- Highlight that this is work in mice, and the translation to humans remains to be seen. Please refrain from presenting the mice data as conclusive evidence of human disease, as this is how the work currently reads.

- Please discuss the differences between the human and mouse liver.

- Please discuss the differences in infection dosages and the consequences. For instance, the wt and type 1 deficient mice have a higher viral load, which could result in earlier detection and better response.

- The conditional KO is performed in adult mice and so cannot be extrapolated to what would happen in neonatal mice. Considering this is the primary motivation for the manuscript, this limitation must be discussed.

Reviewer #2: • E5 expression in scRNA-seq was mostly relegated to Supplementary Materials. Since it is central to correlating infection with IFN-I expression, it should be shown in the main figures.

• Fig. 2B: Abbreviation definitions are unclear, possibly due to formatting.

• Fig. 2C: Expression of B2m and hFcRn appears higher in innate immune cells than hepatocytes. As viral RNAs localized mainly to macrophages in resistant strains (WT, Ifnlr KO) but to hepatocytes in IFN-I–deficient strains, validation of hFcRn expression by IF or flow cytometry would be valuable.

• L255–256: Cxcl10 expression was highest in Ifnar/Ifnlr double KO mice. This requires explanation.

• L257–259: Fig. 3A–C show type I IFNs mainly from DCs in susceptible strains, while in WT only IFN-β is expressed. This does not support the conclusion that four clusters produced IFNs across all genotypes.

• L301–311: Pseudotime analysis suggesting Kupffer cells are not derived from monocytes requires experimental validation (e.g., Ccr2 KO). If not feasible, I recommend removing this section.

• L312: The statement that Mono.mac was the major source of IFN-I is not consistent with Fig. 3A–C and requires clarification.

• L446–451: Sentence repetition.

• L507–509: Please clarify whether neonatal and aged mice show different immune responses in hFcRn(Tg32) background.

• L538: Evidence for Kupffer cell expansion comes only from scRNA-seq. Experimental validation (e.g., Ki67, EdU) is needed.

Reviewer #3: Well written paper.

There is a typo on line 490 'live' should be liver

PLOS authors have the option to publish the peer review history of their article (what does this mean? ). If published, this will include your full peer review and any attached files.

**Do you want your identity to be public for this peer review?** For information about this choice, including consent withdrawal, please see our Privacy Policy .

Reviewer #1: No

Reviewer #2: No

Reviewer #3: No

**Figure resubmission:**

**Reproducibility:**



---

## [Decision Letter · Decision Letter 1]

16 Dec 2025

PPATHOGENS-D-25-02022R1

Tissue-Resident Macrophage and Dendritic Cell Cooperation Drives Type I IFN Immunity to Enteroviruses in the Liver

PLOS Pathogens

Dear Dr. Coyne,

Thank you for submitting your manuscript to PLOS Pathogens. After careful consideration, we feel that it has merit but does not fully meet PLOS Pathogens's publication criteria as it currently stands. Therefore, we invite you to submit a revised version of the manuscript that addresses the points raised during the review process.

We look forward to receiving your revised manuscript.

Kind regards,

Helene Minyi Liu, Ph.D.

Academic Editor

PLOS Pathogens

Michael Letko

Section Editor

PLOS Pathogens

Sumita Bhaduri-McIntosh

Editor-in-Chief

PLOS Pathogens

orcid.org/0000-0003-2946-9497

Michael Malim

Editor-in-Chief

PLOS Pathogens

orcid.org/0000-0002-7699-2064

**Additional Editor Comments:**

Please revise the text according to the reviewers' comments. Clarification of the pseudotime analysis is suggested. Missing punctuations (Line 133) and references (Line 195 mm10 and Line 198/823 UMI).

**Journal Requirements:**

1) Thank you for stating that "All raw data related to sequencing have been deposited and the accession number provided in the text." Please update your Data Availability in the online submission to include the repository name, and the DOI/accession number of each dataset OR a direct link to access each dataset. If your manuscript is accepted for publication, you will be asked to provide these details on a very short timeline. We therefore suggest that you provide this information now, though we will not hold up the peer review process if you are unable.

**Reviewers' Comments:**

Reviewer's Responses to Questions

**Part I - Summary**

Reviewer #1: The authors have extensively addressed my comments from the first review round. I am satisfied with the response and additional experiments provided in the revised manuscript.

Reviewer #2: The revised manuscript addresses several of the initial concerns. However, a number of important issues remain and require further clarification or modification.

Reviewer #3: The authors have addressed my critiques.

**Part II – Major Issues: Key Experiments Required for Acceptance**

Reviewer #1: (No Response)

Reviewer #2: 1. Clarification of the cellular source of IFN-I

The newly added section (lines 606–611) focuses primarily on IFN-I–responsive cells rather than the cellular sources of IFN-I, which was the central issue raised previously. Although additional mouse crosses are time-consuming, further experimental support is necessary if the manuscript is to assert mechanistic interactions between Kupffer cells and dendritic cells in driving IFN-I responses.

2. Colocalization of viral RNA in Kupffer cells

The inclusion of new imaging data is appreciated. However, the colocalization between viral RNA signals and Kupffer cell markers remains difficult to interpret with confidence. Higher-resolution confocal microscopy in single xy-plane images and quantitative colocalization analysis would significantly strengthen this conclusion.

Reviewer #3: NA

**Part III – Minor Issues: Editorial and Data Presentation Modifications**

Reviewer #1: (No Response)

Reviewer #2: 1. Strength of conclusions regarding Kupffer cell–derived IFN-I and Kupffer cell expansion

While I appreciate the technical challenges associated with generating additional genetic models to directly assess IFN-I production by Kupffer cells, and likewise the difficulty of evaluating Kupffer cell expansion during echovirus infection, the current data do not provide strong support for the conclusions presented. I recommend tempering these claims, particularly the statements in lines 612–613 of the Discussion.

2. Interpretation of pseudotime analysis

In the pseudotime section (lines 353–364, R1 version), the manuscript states that Kupffer cells and inflammatory macrophages appear to be distinct but related monocyte-derived populations. Despite this, the subsequent conclusion (lines 362–364) that “the expansion of Kupffer cells during infection is not the result of infiltrating inflammatory macrophages repopulating the resident Kupffer cell niche” is stronger than the data support. As written, this conclusion is potentially misleading and should be revised to align with the evidence.

3. Rationale for focusing on the Mono.mac cluster

The manuscript notes that Mono.mac is “one of three” clusters that express substantial levels of type I IFNs. However, the subsequent analysis (line 365) emphasizes Mono.mac specifically, without explaining why it is highlighted over the other two clusters. Clarification of the rationale for focusing on this cluster is needed.

Reviewer #3: Line 326. Figure 3. The legend indicates L-M. However, Fig. 3 has A-N panels. This should read L-N.

PLOS authors have the option to publish the peer review history of their article (what does this mean? ). If published, this will include your full peer review and any attached files.

**Do you want your identity to be public for this peer review?** For information about this choice, including consent withdrawal, please see our Privacy Policy .

Reviewer #1: No

Reviewer #2: No

Reviewer #3: No

**Figure resubmission:**
---

## [Editor Report · Decision Letter 2]

12 Jan 2026

Dear Dr. Coyne,

We are pleased to inform you that your manuscript 'Tissue-Resident Macrophage and Dendritic Cells Drive Type I IFN Immunity to Enteroviruses in the Liver' has been provisionally accepted for publication in PLOS Pathogens.

Before your manuscript can be formally accepted you will need to complete some formatting changes, which you will receive in a follow up email. The editor still found missing punctuations (Line 133) and references (Line 195 mm10 and Line 198/823 UMI). Please proof-read carefully. A member of our team will be in touch with a set of requests.

Best regards,

Helene Minyi Liu, Ph.D.

Academic Editor

PLOS Pathogens

Michael Letko

Section Editor

PLOS Pathogens

Sumita Bhaduri-McIntosh

Editor-in-Chief

PLOS Pathogens

orcid.org/0000-0003-2946-9497

Michael Malim

Editor-in-Chief

PLOS Pathogens

orcid.org/0000-0002-7699-2064
---

## [Editor Report · Acceptance letter]

Dear Dr. Coyne,

We are delighted to inform you that your manuscript, "Tissue-Resident Macrophage and Dendritic Cells Drive Type I IFN Immunity to Enteroviruses in the Liver," has been formally accepted for publication in PLOS Pathogens.

Best regards,

Sumita Bhaduri-McIntosh

Editor-in-Chief

PLOS Pathogens

orcid.org/0000-0003-2946-9497

Michael Malim

Editor-in-Chief

PLOS Pathogens

orcid.org/0000-0002-7699-2064